

# DISCOL experiment revisited: Assessing the temporal scale of deep-sea mining impacts on sediment biogeochemistry

Laura Haffert[1], Matthias Haeckel[1], Henko de Stigter[2] and Felix Janßen[3]

[1] GEOMAR Helmholtz Centre for Ocean Research Kiel, Wischhofstrasse 1-3, 24148 Kiel, Germany

[2] NIOZ - Royal Netherlands Institute for Sea Research and Utrecht University, P.O. Box 59, 1790 AB Den Burg - Texel, The Netherlands

[3] HGF MPG Group for Deep Sea Ecology and Technology at the Max Planck Institute for marine Microbiology, Bremen and Alfred Wegener Institute Helmholtz Centre for Polar and Marine Research, Bremerhaven, Germany Max-Planck Institute for marine Microbiology, Bremen, Germany

*Correspondence to*: Laura Haffert (lhaffert@geomar.de)

**Abstract.** Deep-sea mining for polymetallic nodules is expected to have severe environmental impacts because in addition to the nodules, benthic fauna as well as the upper reactive sediment layer is removed through the

mining operation, and blanketed by resettling material from the suspended sediment plume. This study aims to provide a holistic assessment of the biogeochemical recovery after a disturbance event by applying prognostic simulations based on an updated diagenetic background model and validated with novel (micro)-biological data. It was found that the recovery strongly depends on the impact type; complete removal of the reactive surface sediment reduces seafloor nutrient fluxes over centuries, while geochemical processes after resuspension and

mixing of the surface sediment are near pre-impact state one year after the disturbance. Furthermore, the geochemical impact in the DISCOL area would be mitigated to some degree by a clay-bound Fe(II)-reaction layer, impeding the downward diffusion of oxygen, thus stabilizing the redox zonation of the sediment during transient post-impact recovery. The interdisciplinary (geochemical, numerical and biological) approach highlights the closely linked nature of benthic ecosystem functions, e.g. through bioturbation, microbial biomass

and nutrient fluxes, which is also of great importance for the system recovery.

## 1 Introduction

This work is part of the JPI Ocean MiningImpact project, which is an integrated research project assessing the potential ecological impact caused by dee-sea mining for polymetallic nodules. Most of these nodules occur in the deep ocean basins at water depths greater than 4000 meter, which represents a part of the ocean that is

largely unstudied. In fact, until recently it was assumed that life would be sparse in the deep ocean where, in the absence of light, the only energy source is falling organic matter produced in the photic zone. Increased research interest and technological advances to overcome the large distances and pressure differences of over 400 times the atmospheric pressure have revealed that the deep-sea hosts a surprising diversity of life forms. Information on benthic ecosystem functions, including spatial heterogeneity, temporal variability, and biogeochemical

feedbacks are still limited, but are an important prerequisite in the evaluation of the environmental impacts of deep-sea mining.



This work focuses on a unique study area in the Peru basin (Figure 1), where in 1989 a circular area with a diameter of about 2 nautical miles was intensely disturbed with a plough-harrow as part of the DISturbance and recolonization experiment (DISCOL) (Thiel and Schriever, 1990). The DISCOL experimental area (DEA) was

revisited before and several times after the initial disturbance, with research concentrating mainly on the characterization and distribution of benthic fauna. The current MiningImpact project makes use of the technological advances that have taken place in recent years, e.g. precise underwater positioning and in-situ experimentation by ROV, in situ oxygen measurement, precise sampling with TV-guided or ROV manipulated equipment and analysis of microbial functions, and attempts to derive a holistic understanding of the deep-sea

mining impact on the closely linked ecosystem functions. The specific objectives of this work are:

(i) Creating a biogeochemical (diagenetic) reference model for the DEA region and validating it based on a comprehensive state-of-the-art geochemical dataset (pore water and solid phase) on both undisturbed and disturbed sediments.

(ii) Determining short to long-term impacts of a deep-sea mining event on the geochemical system
through field data supported simulations.

(iii) Advancing our understanding on the intricate interplay between ecosystem functions and geochemical processes in the context of a benthic disturbance in the deep sea.

**1.1 Biogeochemistry of the Peru Basin**

In comparison to other abyssal areas, the Peru basin receives untypically high organic matter input, which is
fueled by the equatorial high-productivity zone (Weber et al., 2000). The degradation of deposited organic matter plays a crucial role in defining the benthic biogeochemical system (Froelich et al., 1979). The reactions utilize different terminal electron acceptors in the order of decreasing free-energy production, namely oxygen, nitrate, manganese oxide, iron oxide and sulphate and their availability positions the various redox zones in the sediment column. König et al. (1999), König et al. (2001) and Haeckel et al. (2001) have previously identified
and quantified through diagenetic modelling the main biogeochemical processes in the Peru Basin, including the DEA region. They found that organic matter degradation progresses through the stages of oxic respiration, denitrification and manganese reduction before the reactive fraction of organic matter is diminished leaving only the most refractory fraction of organic material to be permanently buried. A large quantity of organic matter is degraded by oxic respiration in a surface 'reaction layer' which is bounded by the bioturbation depth at the
lower end. Oxygen is depleted at a sediment depth of 5 to 15 cm, with the DEA region being on the deeper end. Larger variations were found among the nitrate profiles. In the slightly shallower region of the Peru Basin seawater-derived nitrate and nitrate from oxic respiration is consumed within the bioturbated upper 20 cm of sediment. In contrast, in the DEA region nitrate escapes the reaction layer and diffuses to a depth of about 2 m, where a Fe(II)-rich reaction layer impedes further downward migration (König et al., 2001; see also Figure 2).
This reaction layer was formed during glacial times, when organic matter deposition was strongly increased and potential electron acceptors such as oxygen, nitrate and manganese oxide were exhausted closely below the sediment surface. During these times, structural Fe(III) within smectite lattices acted as the thermodynamically favored electron acceptor, creating Fe(II)-rich surface sediments that were subsequently buried (Lyle, 1983). This process was reversed when waning organic matter input during interglacial periods allowed oxygen and
nitrate to migrate downwards thus creating a 'burn-down' situation of the redox sensitive Fe(II)-rich smectite phase (Figure 2). The position of the reaction front is easily distinguished in deep sediment cores as a transition





from a tan to green color change (Lyle, 1983) and has also been confirmed by Mössbauer spectroscopic measurements (Drodt et al., 1997;König et al., 1997;König et al., 1999).

Comparatively little is known about the redox potential of Fe(II) on different crystallographic lattice positions (coordination sites) in clay minerals, such as nontronite. In particular, information is lacking on the reactivity of the Fe(II) on these sites with respect to oxidation by nitrate and oxygen. Available Mößbauer data (König et al., 2001) from the DISCOL area indicates a refractory fraction of Fe(II) that prevails in the nitrate reduction zone (~10 % of total iron). We propose that part of this refractory fraction is in fact semi-labile and that oxygen, which has a higher redox potential than nitrate, can oxidise this semi-labile Fe(II) and thus presents a second reaction front impeding the downward migration of oxygen.

## 2 Methods

### 2.1 Disturbance experiments

The current study presents information collected in the DEA during the two legs of the SO242 research cruise in 2015. The DISCOL experiment was designed to artificially disturb the surface sediment layer and remove nodules from the surface with a specially designed device, the so-called plough-harrow (Thiel and Schriever, 1990); To this end the ~4150 m deep circular DEA (11 km$^2$) was crossed 78 times starting from various directions resulting in a heavily disturbed area in the center and less disturbed peripheral regions. In total about 20% of the DEA was directly ploughed with the remaining 80% being not directly disturbed but – at least in part – covered by an up to 30 mm thick re-settled sediment blanket (Schriever and Thiel, 1992).

During the first leg of SO242 an additional artificial disturbance was created using an epibenthic sled (EBS) which was sampled 5 weeks later on the second leg of SO242. The epibenthic sled (CliSAP-Sled; Brenke (2005)) has a width of 2.4 m and weight of 880 kg (in air). Unlike the plough-harrow, which 'mixes' the upper sediment layer, the EBS largely scrapes off the upper centimetres of sediment (most of the reactive, bioturbated layer), including embedded nodules and bulldozes them sideways from the track.

### 2.2 Sampling procedure

A map of the sampling stations as well as a table of the exact positions are provided in Figure 1 and Table 1, respectively. Several devices were deployed at each sampling site to retrieve surface and subsurface sediment samples (Greinert, 2015;Boetius, 2015): (i) a multiple-corer (MUC, Oktopus, Kiel, Germany), which samples the upper 20 – 40 cm of the sediment including the overlying bottom water; The corer was equipped with a TV-Camera system for precise positioning, e.g. for sampling specific seafloor features like the disturbance tracks; (ii) a ROV-deployed push-corer (PUC) focusing on small-scale topographic features created by the seafloor disturbance; (iii) a box-corer (BC - USNEL type; Hessler and Jumars, 1974) to sample the sediment including macrofauna and nodules; and (iv) a gravity-corer (GC) extracting sediment up to 10 mbsf.

After core retrieval the samples were brought into the ship's cold room (approx. 4 °C). 1 m long sections of the gravity cores were cut into two half cylinders dedicated for sampling and archiving. The working half of the gravity cores were sampled by extracting 3 cm thick slices at intervals of 20 to 40 cm. Sediment from the plastic liners of the other corers were extruded with a piston and cut into 0.5 to 2 cm thick slices under an oxygen-free, argon atmosphere in a glove bag. Subsequently, the pore water was extracted using a low pressure-squeezer (argon gas at 3-5 bar) and filtered through a Nuclepore 0.2 µm polycarbonate filter. Porewater extracted from



the samples was stored in two recipient vessels: (i) an acidified sample (i.e. pH<1 with 20 µl 30% HCl suprapur) for analysis of metal cations and (ii) a non-acidified sample for nutrients. Additional sediment samples were collected for porosity, organic carbon content and radiometric analysis.

### 2.3 Analytical methods

Analyses for the porewater solutes $NO_3^-$, $NO_2^-$, $NH_4^+$, $PO_4^{3-}$, $SiO_4^{4-}$ and $Fe^{2+}$ were completed onboard using a

Hitachi UV/VIS spectrophotometer. The respective chemical analytics followed standard procedures (Grasshoff et al., 1999), i.e. nitrite and nitrate (after reduction with Cd) were measured as sulphanile-naphthylamide, ammonium was measured as indophenol blue, phosphate and silicate as molybdenum blue, and iron with ferrospectral. The total alkalinity of the porewater was determined by titration with 0.02 N HCl using a mixture of methyl red and methylene blue as indicator. The titration vessel was bubbled with argon to strip any CO2

produced during the titration. The IAPSO seawater standard was used for calibration. Further details on analytical methods, e.g. analytical precision and accuracy, are given in Table 2.

At the NIOZ laboratory sediment profiles of $^{210}$Pb and $^{226}$Ra, radio-isotopes from the $^{234}$U decay series were measured in multicores for determining biological mixing rates integrated over a ~100 year time-scale. Total $^{210}$Pb and $^{226}$Ra activity were determined directly by gamma-spectrometry, whilst total $^{210}$Pb was also measured

indirectly by alpha-spectrometry via its granddaughter isotope $^{210}$Po. Activities of anthropogenic $^{137}$Cs proved to be generally below detection level. Radionuclide activities are reported in the data base as mBq g$^{-1}$ dry sediment.

For gamma-spectrometry, a few grams of freeze-dried and homogenized sediment sample was contained in a 5 cm diameter plastic petri dish, which was closed with tape and sealed gas-tight in a plastic envelope. After

leaving the sample for at least 4 weeks to ensure equilibrium, measurement of $^{210}$Pb and $^{226}$Ra were undertaken with a Canberra Broad Energy Range High Purity Germanium Detector (BEGe), using the 46.5 keV line for $^{210}$Pb and 295.2, 351.9 and 609.3 keV lines for $^{226}$Ra. The detector, connected to a computer via a Digital Spectrum Analyser (DSA-1000), counted the radionuclide activities with Genie 2000 gamma spectroscopy software. The detector was externally calibrated with a Geological Certified Reference Material IAEA/RGU-1,

with reference date of 01-01-1988. A monitor standard IAEA-300 provided quality control. Excess $^{210}$Pb activities were calculated by subtracting $^{226}$Ra activity averaged for the 295.2, 351.9 and 609.3 keV lines from the measured total $^{210}$Pb activity.

For alpha spectrometry measurement of $^{210}$Po, 0.5 g of freeze-dried and homogenised sediment sample was spiked with 1 ml of a standard solution of $^{209}$Po in 2M HCl, and then leached for 6 h in 10 ml of concentrated

HCl heated to 85°C. After diluting the fluid with 45 ml of demineralised water and adding 5 ml of an aqueous solution of ascorbic acid (40 g L$^{-1}$), natural $^{210}$Po and added $^{209}$Po were collected from the fluid by spontaneous electrochemical deposition on silver plates. For subsequent alpha-spectrometry, Canberra Passivated Implanted Planar Silicon detectors were used. $^{210}$Pb activity was calculated from $^{210}$Po, assuming secular equilibrium and correcting for the time elapsed since collection of the samples.

### 2.4 Numerical methods

We follow the classical approach in early diagenetic modeling where partial differential equations represent the diffusive and advective fluxes coupled to a reaction term ($R$), as formulated by Berner (1980). When applying a conversion via the volume fraction, expressions for solutes and solids can be derived, respectively:





$$\frac{\partial \phi C_{pw}}{\partial t} = \frac{\partial}{\partial x} \left( \phi D \frac{\partial C_{pw}}{\partial x} - \phi\, u\, C_{pw} \right) - \phi R\left( C_{pw} \right)$$

and

$$\frac{\partial (1-\phi) C_s}{\partial t} = \frac{\partial}{\partial x} \left( (1-\phi) Db \frac{\partial C_s}{\partial x} - (1-\phi)\, w\, C_s \right) - (1-\phi) R(C_s)$$

where $t$ and $x$ represent time and the depth under the seafloor, respectively; $C_{pw}$ and $C_s$ are the molar species

concentration of solute or solid, respectively; $\phi$ is the porosity; $u$ and $w$ represent fluid and sediment velocity

due to burial. Solute diffusion and sediment bioturbation is scaled by the effective diffusion coefficient $D$ and

the bioturbation coefficient $Db$.

The porosity-depth profile is assumed to be produced by steady-state compaction (Berner, 1980) and is

approximated empirically by the following exponential function (Murray et al., 1978;Berner, 1980;Martin et al.,

1991;Rabouille and Gaillard, 1991a, b):

$$\phi(x) = \phi_\infty + (\phi_0 - \phi_\infty)\, e^{-\beta x}$$

where $\phi_\infty$ is the porosity at infinite depth, $\phi_0$ is the porosity at the sediment surface (x=0) and $\beta$ is the porosity

attenuation coefficient.

The fluid and sediment velocity-depth distribution is calculated according to the mathematical convention of

Luff and Wallmann (2003):

$$u(x) = \frac{\phi_\infty w_\infty}{\phi(x)} \text{ and } w(x) = \frac{1-\phi_\infty}{1-\phi(x)}\, w_\infty$$

where $w_\infty$ is the burial velocity at infinite depth.

The effective diffusion coefficient was tortuosity corrected:

$$D(x) = \frac{D^o(x)}{\theta^2(x)}$$

with $D_0$ being the infinite dilution molecular diffusion coefficient calculated after Berner (1980) and $\theta^2$ being

the squared tortuosity. The tortuosity can be related to the porosity $\phi$ according to Boudreau (1996):

$$\theta^2(x) = 1 - 2\ln\phi(x)$$

The bioturbation profile is approximated by the arbitrary function

$$Db = Db^0 \cdot 0.5\, erfc\left( \frac{x - x_{Db}}{\beta_{Db}} \right)$$

Where $Db^0$ represents the maximum bioturbation intensity at the sediment surface, $x_{Db}$ is the bioturbation half

depth and $\beta_{Db}$ the bioturbation attenuation coefficient.

The diagenetic equations consisting of a set of partial differential equations, are solved via a finite difference

scheme (1D uneven grid) and subsequent minimization of the ordinary differential equations by inbuilt ODE

solvers of MATLAB.

The reaction network shaping the geochemistry of the upper sediment metres, namely organic matter

degradation and secondary redox reactions including the oxidation of ammonium, dissolved manganese and

refractory clay bound Fe(II) are listed in Table 3 including the corresponding rate expressions. The diagenetic

model is applied to define a background or reference geochemical system. This is then used to derive transient

prognostic models in response to different impact types. A detailed parameterization of the background and

prognostic models is provided in Table 4.





The bioturbation coefficient $b^0$, a critical input parameter to the background model, is derived by fitting a
simplified version of the background model to summed [210]Pb, [226]Ra and [230]Th radiometric data. The physical set
up of the model is the same as the background model with the difference that only the above mentioned
radiometric species and their first order radioactive decay is considered.

## 3 Results

### 3.1 Geochemistry of the DEA region

The parameter depth profiles for each core are presented in Figure 3 – 6, grouped in 'outside DEA' and 'inside
DEA' stations for the gravity cores and for the multiple and box cores in reference sites and various types of
disturbed sites.

Geochemical trends observed in the sediment cores from reference stations agree in general well with those
observed in the same area during a previous investigation (Haeckel et al., 2001). The upper reactive sediment
section (up to ~20 cm.b.s.f) is markedly different from the deeper sediment sections (up to 10 m). It is
characterized by a sharp decrease in porosity (from 0.94 to approx. 0.86, Table 4) and is rich in manganese
oxides (Paul et al., under review), which gives this layer its distinct dark brown color. Below this layer, the
sediment takes on a light brown to grey brown (tan) color, which is replaced by a grayish green color - the shift
to Fe(II)-rich lattices in the smectite phase – at a depth of 2 – 2.5 m (38GC1, 84GC3, 100GC5, and 123GC6,
Table 1). Organic carbon content in the DEA region oscillates about a mean value of 0.5 to 0.75 wt% in the
upper 50 cm, which is controlled by the difference in organic carbon input during glacial (higher $C_{org}$ content)
and interglacial (lower $C_{org}$ content) sedimentation regimes (Haeckel et al., 2001). With depth the organic
carbon content decreases steadily to about 1 wt% at 10 m.b.s.f. Contrary to previous observations that nodules in
the DEA area occur only at the sediment surface, investigation of the current set of gravity cores revealed buried
manganese nodules at depths of 4 to 8 m. The undisturbed appearance of the sediment surrounding these buried
nodules, and evidence of diagenetic alteration of the nodules, proves that these nodules were genuinely buried
and do not represent a coring artifact.

In situ oxygen profiles confirm that the upper brown layer represents the oxygenated zone (Figure 6). In this
zone, oxic respiration of organic matter not only consumes downward diffusing oxygen but increases
ammonium and, via nitrification, also nitrate (Figure 4). While the trend in the nitrate profiles is clearly
discernable, ammonium concentrations remain rather low (10 – 20 µmol/l) and are scattered and are likely
controlled by secondary processes, such as adsorption and desorption (Haeckel et al., 2001). Below the oxic
dark brown layer, denitrification commences and nitrate declines within the upper 2 -3 m of the sediment and is
entirely depleted in the cores with the greyish green Fe(II)-rich reaction layer (51GC2, 84GC3, 100GC5,
123GC6, Figure 3). The depleted nitrate profiles tend to be more linear (e.g. 84GC3) than the profiles of cores
where the Fe(II)-rich layer is absent (e.g. 132GC7), as is typical in the presence of a reaction layer (Goloway
and Bender, 1982;Wilson et al., 1985;Jahnke et al., 1989). Dissolved manganese is strongly redox sensitive and
is absent in the upper oxic zone. Below the oxic zone, organic matter degradation progresses to manganese
oxide reduction and produces dissolved manganese, which increases steadily until it reaches an asymptotic
concentration at 6 m depth with terminal concentrations ranging between 25 and 120 µmol/l. Diagenetic
simulations (Haeckel et al., 2001) could not reproduce this strong increase in dissolved manganese and the





discovery of buried manganese nodules now confirms that the dissolved manganese concentrations are indeed additionally shaped by the dissolution of manganese nodules at a depth of 4 – 8 m.

Alkalinity profiles are largely unchanged over the entire sampling domain (up to 10 m), but display some
variations in the upper 5 cm. Haeckel et al. (2001) have argued that the alkalinity profile is likely influenced by carbonate geochemistry, so we will neglect the upper alkalinity profile in our interpretations. Dissolved iron concentrations were near or below detection limit in all cores and sulphate concentrations remained close to their sea water value, which is in line with the absence of iron and sulfate reduction in the Peru basin.

### 3.2 Diagenetic reference model

With the MiningImpact sampling campaign specifically targeting the DEA region, we were able to update the parameterization of the previous model presented by Haeckel et al. (2001). The main differences are: (i) The presented model applies a 3G-model, i.e. allows for three fractions of organic matter with varying reactivity, and thus a better fit to the field data. And (ii) The bioturbation intensity ($D_B = 0.65$) and depth profile are now based on radioactive disequilibria between radionuclides of the natural uranium and thorium series in the DEA region
(Figure 7). The numerical simulations predict that excess $^{210}$Pb (difference between solid and dashed lines) exists down to ~10 cm and below the measured activity is supported by $^{230}$Th (dotted line) and $^{226}$Ra (difference between dashed and dotted lines). A relatively high scatter of radioactive activities of 1.5-3 Bq/g was observed in the uppermost sediment samples (0-0.5 cm) reflecting lateral inhomogeneities, These could be due to sediment focusing created by local topography, e.g. small depressions acting as deposition centers for fluffy
material or by 'hotspots' of non-local mixing by individual organisms.

A satisfactory background model fit could be achieved for all relevant parameters with particular focus being placed on accurately simulating the redox zonation of the sediment, especially the oxygen penetration depth and the concurrent increase in dissolved manganese (Figure 8). It should be noted that the geochemical state of the Peru basin is likely transient in nature arising from long-term (glacial/interglacial) as well as short-term (ENSO
time scale) variations in the depositional flux of organic matter (König et al., 2001). Similar to the downward migrating nitrate front, it is likely that the pore water oxygen content also increases with currently waning organic matter input. The analytical oxygen profiles display less curvature (more linearity) than the modelled profiles, which suggests that secondary reactions, such as a semi-labile Fe(II)-oxygen reaction front, additionally shape the oxygen profile. However, due to the uncertainty in a transient parameterization, we model
the reference sediments as a steady state scenario - justified by the generally good agreement of the model with the geochemical data.

### 3.3 Impact characterization

All sample stations were affected by at least one of the following processes: (i) removal of the upper dark brown reactive layer, (ii) mixing and redistribution of the surface sediment resulting in piling up ('riges') and removal
('furrows'), (iii) turning of the upper sediment upside down or bringing up deep sediment to the surface – both leading to exposal of sediment from below the reactive layer at the surface and (iv) deposition of re-suspended sediment. Furthermore, there were distinct differences in the physical impact of the plough-harrow and EBS track. While the EBS scraped off the surface sediments and relocated them to the side of the track, the plough-harrow mixed and displaced sediments resulting in a variety of disturbance structures. In the following a



detailed description of the physical and geochemical impact is provided and will be complemented by relevant information on the microbiological impact from a parallel study (Vonnahme et al., submitted).

Sediments inside the EBS track are largely devoid of the upper reactive brown layer, exposing the tan-colored subsurface sediment in places. The surface of the track is smooth indicating that sediment mixing is negligible inside the tracks. Removal of the upper reactive layer resulted in decreased sediment porosity by exposing more compacted sediment and brought the boundary below which dissolved manganese is found in the sediment pore water as close as 5 cm bsf. (Figure 4 and 5). In contrast, the oxygen penetration depth does not vary significantly from the reference area, the profile shape is however slightly more linear (Figure 6). In line with the severe disturbance of the sediment, signs of bioturbation are lacking and the microbial cell count in the surface sediment is reduced by 50% with the microbial communities being significantly different from the reference sites (Vonnahme et al., submitted).

The disturbance of the surface sediment layer brought about during the DISCOL experiment were still clearly visible after 26 years and the TV-guided sampling campaign was able to specifically target the following features: (i) The area next to the plough-harrow track ('outside track'), that is not mechanically disturbed but affected by resettling sediment from the disturbance plume. Geochemical profiles do not indicate a significant difference to the reference site (Figure 4), especially with the natural heterogeneity of the area in mind. Differences in the microbial functions compared to the reference stations are also within the methods' confidence level and recent bioturbation confirms that the sediment is presently colonized by burying megafauna (Vonnahme et al., submitted). (ii) The furrows drawn in the seabed by the plough-harrow show some distinct differences to the reference sites. The reactive dark brown surface layer is reduced in thickness and is overlain by re-settling sediment which preferentially accumulated in the depression. While geochemical profiles do not differ significantly from the reference site, biological processes are noticeably decreased (Vonnahme et al., submitted). (iii) 'Ridge' sediments show the same disturbance effects, albeit more severe, and seafloor integrity is compromised by deep cracks throughout the sediment (Vonnahme et al., submitted). (iv) Exposed subsurface sediment observable as lighter-coloured patches show the highest level of disturbance, similar in character to the fresh surface created by the EBS. The reactive brown layer is absent and thus the porosity of the surface sediment reduced (Figure 5). Biological processes are significantly impacted. Bioturbation channels are scarce and disconnected with the surface, indicating a pre-impact origin. Microbial cell numbers are reduced by 30% with the microbial communities being more similar to those of the subsurface sediment (14-16 cm.b.s.f.) of the reference stations than those of the surface sediments (Vonnahme et al., submitted).

**3.4 Prognostic simulations**

The following presentation of simulation results focuses on the distribution of oxygen, which plays a central role in biogeochemical processes: Oxygen is essential to all multicellular life and represents the electron acceptor with the highest energy yield. It exerts an important control on the sediment redox zonation and associated reactions (most importantly the oxic respiration of organic matter), which in turn control the distribution of other nutrients, such as ammonium and nitrate. Different impact types were simulated by two 'end member' scenarios. The first scenario assumes that 10 cm of the upper sediment were removed, i.e. the first 10 cm of the reference steady-state profiles were discarded, exposing anoxic subsurface sediments to bottom water concentrations. The second scenario floods the upper 10 cm of the sediment with oxygen, representing, on one hand, the effect of sediment mixing and cracking and, on the other, the effect of re-suspension and subsequent



settling of freshly oxygenated reactive sediment (Figure 9). Impact simulations use the steady state background model as initial profiles (adjusted for the modelled impact type, i.e. removing the upper 10 cm or setting oxygen concentrations to bottom water values) and augmented by an additional Fe(II) profile - an oxygen reaction layer (Figure 2) - as justified by the surprisingly constant oxygen penetration depth at the reference and all disturbed sites (Figure 6). This Fe(II) reaction layer is crucial in the prognostic simulations which would otherwise predict

that oxygen rapidly diffuses into deeper sediments after a disturbance event (Figure 10 and 11). It is assumed that bioturbation is inhibited immediately after the impact with a linear increase to undisturbed reference bioturbation intensity within 100 to 200 years.

The transient models showed that the impact of the removal of the reactive upper 10 cm layer is much more severe and longer lasting compared to the 'oxygen flooding' scenario. And simulations show no distinguishable

difference if 8 or 12 cm are removed instead of 10. This is because the greatest disturbance effect is caused by the absence of the labile organic matter fraction, which is restricted mostly to the upper centimetres of the sediment.

*Short term impact.* Immediately after the removal of the highly reactive surface sediment, solute profiles are in strong disequilibrium and mostly diffusion controlled. This is especially true for oxygen, which rapidly diffuses

into previously anoxic sediments. The Fe(II) layer plays an insignificant role in the initial post-impact weeks but commences to inhibit further penetration of the oxygen front at 5 weeks. In the absence of the reactive surface layer, organic matter degradation rates, which strongly shape the reference oxygen profile, play only a minimal role in the weeks after the disturbance. The modelled 'removal' profiles agree well with the analytical in situ oxygen profiles of the 5 week old EBS track and confirm that the linear nature of the analytical profiles is likely

shaped by a mixture of a downward diffusing oxygen front and the initiation of the oxygen-Fe(II) reaction layer. For comparison, impact simulations assuming full recovery of the bioturbation intensity by 200 years instead of 100 years, are also shown. Here, the oxygen profile is also distinctly more linear but even more reduced sediment mixing through bioturbation allows oxygen to penetrate slightly deeper (18 cm.b.s.f.) compared to the in situ oxygen profiles (13 cm) and to the 100 year bioturbation recovery interval (10 cm.b.s.f, Figure 10).

The 'oxygen flooding' scenario produces a very different oxygen profile. It is shaped mainly by the reaction of the anomalous oxygen with the reactive (labile) organic matter, especially in the upper centimeters where the labile organic matter is most concentrated. The difference to the EBS track in situ profiles supports the observation that the EBS track is mainly affected by sediment removal, rather than mixing (Figure 11).

*Medium term impact.* The existence of the Fe(II)-oxygen reaction layer plays a particularly important role in the

decades after the disturbance. Oxygen is less efficiently removed from the pore water in both scenarios: Removal of the reactive layer significantly reduces oxygen uptake through organic matter degradation and the reactivity of the organic matter was reduced due to reaction with the anomalous flooding with oxygen. In the absence of an oxygen impeding reaction layer, oxygen penetration depths would consequently be significantly increased in both cases (Figure 10 and 11). Simulating the 26 year old DISCOL disturbance has shown that the

existence of an oxygen consuming reaction layer is a necessary prerequisite. The 'removal scenario' produces slightly more linear profiles compared to the 'oxygen flooding' simulation because of the relative importance of the Fe(II)-oxygen reaction layer over organic matter degradation process. The slight misfit of both scenarios with the in situ oxygen data suggests that the DEA tracks are most likely affected by a mixture of both scenarios, possible leveled out by lateral diffusion effects. Ongoing organic matter flux at the surface sediment





builds up a new post-impact reactive layer with reactivities higher than in the reference model due to the
        ineffective mixing into deeper sediments through bioturbation (Figure 10 and 11). Thus, differences between
        bioturbation recovery scenarios, i.e. 100 and 200 years, increase over time with an increasing amount of fresh
        organic matter that can potentially be mixed into the sediment.

        *Long term impact.* It takes centuries for the geochemical processes to completely recover after the removal of
the upper reactive sediment. While the observed geochemical profiles of the disturbed sites show surprisingly
        little variation compared to the reference site, overall surface fluxes vary significantly during the first 100 years
        after the impact. This is especially true for the oxygen flux into the sediment, which is more than halved within
        the first year of the impact (Figure 10). Within the first decades after the impact the build-up of the labile
        organic matter, and its mixing into deeper sediment through bioturbation, gains influence. With increasing
depletion of oxygen through organic matter degradation, oxygen concentrations decrease alongside with Fe(II)-
        oxygen reaction rates. The transition from Fe(II)-controlled to organic matter-controlled oxygen profiles is
        accompanied by a slight increase in the oxygen penetration depth (Figure 10). The exact quantification of this
        transition is difficult to estimate, but the fact that 26 years after the impact oxygen profiles are still efficiently
        impeded by the Fe(II)-oxygen reaction layer suggests that this process will significantly reduce the oxygen
penetration depth until the organic matter degradation rates are near pre-disturbance levels (ca. 100 years after
        the impact). Interestingly, while steady-state biogeochemical processes strongly depend on the depth and
        intensity of bioturbation (Haeckel et al., 2001), the time frame of the bioturbation recovery has very little
        influence on the overall impact recovery process (Figure 10). This is due to the fact, that the limiting factor is
        the availability of labile organic matter, which is replenished on a much longer time scale compared to the
faunal recovery (Stratmann et al., 2018).

        Recovery of the sediment that was affected by the 'oxygen flooding' scenario is occurring within 100 years and
        is already in a near pre-impact state 1 year after the disturbance. Here, the Fe(II)-oxygen reaction layer is
        predominantly responsible for restricting the anomalously high oxygen levels within the upper 20 cm of the
        sediment. Organic matter degradation rates remain high throughout the recovery process and build-up of fresh
labile organic matter that replenishes the loss through the introduction of anomalous oxygen concentration
        occurs within the recovery phase of the bioturbation (Figure 11).

        ## 4. Discussion

        The benthic ecosystem is shaped by a balanced interplay of physical, chemical and biological factors, which are
        fueled by the deposition of organic matter, especially its labile fraction (Figure 12): The degradation of organic
matter is microbially catalyzed and in turn provides nutrients to the microbial community. The associated
        microbial growth provides biomass that sustains meio and macrofauna and forms an important component of the
        benthic food web. The concomitant bioturbation of the sediment induces a variety of changes, such as lowering
        the sediment shear strength and the vertical transport of fresh reactive organic matter into deeper sediments
        (bioturbation) through burrowing invertebrates (Krantzberg, 1985, and references therein). The occurrence of
labile organic matter in deeper sediments where oxygen is not as easily replenished compared to surface
        sediments places an important restriction on the oxygen penetration depth (Haeckel et al., 2001). Bioturbation
        plays thus a deciding factor in early-diagenetic, especially redox sensitive, processes. These are typically closely
        linked to the microbial functions and with the introduction of fresh and digested organic matter at depth by



infauna new material for additional degradation and microbial colonization, close the causal cycle. Changes in
any compartment of the ecosystem functions will entail modifications of all biogeochemical processes. A
natural example is the long-term glacial/interglacial or short-term ENSO driven change in the depositional flux
of organic matter to the seafloor imposing gradual cycles in the redox zonation of the sediment (König et al.,
2001).

Mining the seafloor will have an abrupt and direct impact on several benthic ecosystem functions at a time: (i)
through the removal of all seafloor fauna (Vanreusel et al., 2016;Jones et al., 2017;Gollner et al., 2017;Bluhm,
2001;Borowski, 2001), (ii) through changing the physical characteristics of the sediment surface, i.e. by
exposing more compacted sediments with increased shear strength, and removing manganese nodules as hard-
substrate habitats and (iii) through the displacement and removal of labile organic matter.

Biological, geochemical and numerical results of the MiningImpact project significantly improve our
understanding of the interlinked ecosystem recovery processes in the deep-sea. Removal of the upper reactive
surface layer and its fauna will halt bioturbation and organic degradation rates and significantly decrease
microbial activity (Figure 12A and B). Based only on deep-sea typical microbial biomass turnover rates and
conservative bacterial doubling times, recovery of microbial abundances should occur within 2 – 10 years.
Observations show, however, that the microbial activity remained significantly reduced in the decades following
the DISCOL disturbance experiment (Vonnahme et al., submitted). Microbial recovery thus appears controlled
by the re-establishment of the labile organic matter fraction, which is still significantly reduced at this times
scale. Low nutrient fluxes and the downsized microbial biomass cannot sustain pre-impact faunal abundances
(Stratmann et al., 2018), and recolonization is obstructed by the comparatively hard substrate and absence of
hard-substrate habitats (nodule surface) in impacted areas (Vonnahme et al., submitted). This directly affects the
macrofaunal efficiency to mix surface sediments into deeper layers which is presented in the simulations by the
reduced bioturbation coefficient (Figure 12). It should be noted that even if sediment physical conditions and
organic matter availability have recovered, it is unclear whether the pre-impact infauna is still available to
recolonize the disturbed area.

Prognostic simulations showed that the time scale of the ecosystem recovery ultimately depends only on one
factor: the availability of labile organic matter. The amount of reactive organic matter determines the intensity
of the food-web activity (i.e. summed carbon cycling, Stratmann et al. (2018)) and has to reach pre-impact
levels to allow for the interlinked ecosystem to recover. In this context, two aspects should be particularly
emphasized: (i) deep-sea sediments are characterized by extremely low surface sedimentation rates (~0.004
cm/a) and complete recovery of the labile organic matter fraction takes up to 1000 years (Figure 12). In the
Clarion-Clipperton Zone, located also in the abyssal plains of the Pacific Ocean, surface sedimentation rates are
even lower (0.0002 – 0.001 cm/a; Volz et al. (2018)). In this area, where nodule mining is expected to
commence, potential impact recovery after removal of the upper reactive layer was predicted to occur on an
even longer (millennia) time scale (Volz et al., under review). (ii) Ecosystem impact and consequent recovery is
strongly dependent on the type of disturbance, more precisely, on the amount of labile organic matter removed.
Microbial activity at disturbed sites could be directly correlated to an arbitrary disturbance gradient (Vonnahme
et al., submitted), which in turn coincides with the thickness of the reactive dark brown surface sediments that
was removed during the experimental disturbance. These findings are in line with the results of the prognostic
simulations, which showed that the impact is much less severe and recovery times significantly reduced





(decades, Figure 11) if sediments are simply mixed or affected by resettling sediment ('oxygen flooding'
scenario), compared to the removal of the upper reactive layer (centuries, Figure 10 and 12).

**5. Conclusions**

Previous diagenetic models of the DEA (Haeckel et al., 2001) were updated by a field-based bioturbation coefficient and application of a 3G-organic matter model. A near-surface Fe(II)-oxygen reaction layer was introduced in the prognostic simulations, which represents the refractory Fe(II) phase that did not react with the
downward progressing interglacial nitrate front. The surprisingly similar oxygen penetration depths in reference and 26 year old disturbed sediments indicates that this refractory Fe(II) appears to be an efficient barrier for the progressing oxygen front thus defying previous impact simulation that predicted a significant downward migration of the oxygen front (König et al., 2001).

Transient simulation results have significantly enhanced the quality of the interpretation of the observed
geochemical profiles, which on first sight do not exhibit marked differences among the various disturbed and reference sites. The geochemical recovery after a mining related removal of the upper reactive sediment layer can be divided into three stages: (i) the initial diffusion driven equilibration of the post-impact profiles (within weeks of the impact), (ii) burn down of the Fe(II)-oxygen reaction layer (decades after the impact) and (iii) re-establishment of the reactive organic matter layer (centuries after the impact). If the reactive surface sediment is
not removed but instead mixed or temporarily re-suspended, anomalously high oxygen concentrations alter biogeochemical processes on a decadal scale, albeit much less severe.

The interdisciplinary (geochemical, numerical and biological) approach to characterise the impact of benthic disturbances on early diagenetic processes provided valuable information on post-impact processes. On one hand it identifies variables that are suitable as indicators of benthic ecosystem health and also allow
identification of significant adverse change of the benthic environment. Metabolites of microbial activity were found to be particularly sensitive to disturbances and can even resolve grades of impact (Vonnahme et al., submitted). Biogeochemical processes responded more subtle to the disturbance with oxygen, nitrite and bioturbation activity being the most suitable variables. On the other hand, this work highlights the closely linked nature of different benthic ecosystem functions, which is also true during the recovery from mining impacts.
The main factor constraining the time frame of the geochemical recovery is the availability of reactive (labile) organic matter, emphasizing the importance of the impact type (sediment removal versus mixing and resuspension) and also the natural depositional flux of organic matter onto the sediment surface (e.g. DEA versus the Clarion-Clipperton Zone). The microbial degradation of the available reactive organic matter fraction defines the base of the food web structure, which eventually sustains metazoans. Some of the epifauna burrows
into the seafloor (bioturbation) and in turn has a critical influence on geochemical, especially redox sensitive, processes and fluxes. Recovery of the system is thus only possible if the ecosystem functions of all compartments are restored. Our results also show that it is important to identify regional specialties, such as the reactive Fe(II) layer in DISCOL sediments, because they may affect the biogeochemical response to a benthic disturbance drastically.
While biogeochemical fluxes may recover close to the pre-impact state, the nodule ecosystem cannot recover, since the essential hard substrate, the polymetallic nodules, have been removed. Consequently, a new nodule-free ecosystem with very different faunal communities, functions and services has to establish in the mining



areas and its thickly blanketed surrounding. Furthermore, this new ecosystem will take much longer time scales
to establish because it is largely controlled by the recovery of the underlying biogeochemical fluxes and
processes that fuel abyssal life.

## Acknowledgements

The authors would like to thank A. Bleyer, R. Surberg, B. Domeyer and K. Hamann for co-working on sample
collection on RV Sonne and onboard and onshore pore water analyses. P. van Gaever is thanked for carrying out
the radionuclide analysis. We are also indebted to the captain and crew of RV SONNE for their invaluable
support during the cruise SO242. This work was funded by the German Federal Ministry of Education and
Research through the MiningImpact project (grant no. 03F0707A) of the Joint Programming Initiative of
Healthy and Productive Seas and Oceans (JPIO). The authors are solely responsible for the content of this paper.

## Data availability

The geochemical data presented in Figure 3 – 5 are publically available in the PANGAEA database with the
DOI https://doi.org/10.1594/PANGAEA.905377.  The PANGAEA database also hosts the activity
measurements of the $^{210}$Pb series presented Figure 7 with the DOIs  https://doi.org/10.1594/PANGAEA.905442
(alpha spectrometry) and https://doi.org/10.1594/PANGAEA.905443 (gamma spectrometry).

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





**Tables**

**Table 1**. Sampling station locations and description

| Area | Station | Description | Latitude (S) | Longitude (W) | Water Depth / m |
|------|---------|-------------|--------------|---------------|-----------------|
| **Reconnaissance** | | | | | |
| **Outside DEA** | | | | | |
| | | *Southern reference* | | | |
| | 38 GC 1 | southern reference with nodules | 7°07.537' | 88°27.047' | 4161 |
| | 34 MUC 6 | southern reference with nodules | 7°07.524' | 88°27.031' | 4162 |
| | | *Western reference* | | | |
| | 89 GC 4 | western reference with nodules | 7°04.562' | 88°31.577' | 4125 |
| | 80 MUC 22 | western reference with nodules | 7°04.542' | 88°31.581' | 4130 |
| | | *Eastern reference* | | | |
| | 123 GC 6 | eastern reference with(out) nodules | 7°06.045' | 88°24.848' | 4208 |
| | 119 MUC 31 | eastern reference with(out) nodules | 7°06.033' | 88°24.826' | 4204 |
| | | *Volcano crater* | | | |
| | 132 GC 7 | small volcano crater with dense nodules | 7°03.369' | 88°26.031' | 4152 |
| | 129 BC 26 | small volcano crater with dense nodules | 7°03.373' | 88°26.026' | 4144 |
| **Inside DEA** | | | | | |
| | | *DEA plough tracks* | | | |
| | 51 GC 2 | DEA west plough tracks | 7°04.411' | 88°27.836' | 4148 |
| | 56 MUC 12 | DEA west plough tracks | 7°04.414' | 88°27.760' | 4149 |
| | 61 MUC 13 | DEA west outside plough tracks | 7°04.378' | 88°27.781' | 4148 |
| | 70 MUC 17 | DEA west resettled sediment plume | 7°04.400' | 88°27.778' | 4128 |
| | | *DEA black patch* | | | |
| | 84 GC 3 | DEA black patch in sidescan sonar | 7°03.951' | 88°27.093' | 4146 |
| | 74 MUC 20 | DEA black patch in sidescan sonar | 7°03.945' | 88°27.097' | 4150 |
| | | *DEA central/trough* | | | |
| | 100 GC 5 | DEA central | 7°04.342' | 88°27.442' | 4151 |
| | 108 MUC 26 | DEA central | 7°04.483' | 88°26.919' | 4169 |
| **Microhabitats** | | | | | |
| | | *Reference* | | | |
| | 34 MUC 6 | southern reference with nodules | 7°07.524' | 88°27.031' | 4162 |
| | 80 MUC 22 | western reference with nodules | 7°04.542' | 88°31.581' | 4130 |
| | 119 MUC 31 | eastern reference with(out) nodules | 7°06.033' | 88°24.826' | 4204 |
| | | *DEA outside tracks* | | | |
| | 61 MUC 13 | DEA west outside plough track | 7°04.378' | 88°27.781' | 4148 |
| | 146 ROV PC79 | DEA west 20 m off plough track | 7°04.4000' | 88°27.8266' | 4140 |
| | 166 ROV PC70 | DEA east 20 m off plough track | 7°04.4585' | 88°26.9240' | 4143 |
| | 229 MUC | DEA south outside plough track | 7°04.6970' | 88°27.3970' | 4133 |
| | | *DEA track furrow* | | | |
| | 146 ROV PC77 | DEA west plough furrow | 7°04.4110' | 88°27.8363' | 4139 |
| | 166 ROV PC69 | DEA east plough furrow | 7°04.4780' | 88°26.9178' | 4143 |
| | 219 ROV PC75 | DEA south plough furrow | 7°04.6930' | 88°27.4540' | 4155 |
| | | *DEA ridge* | | | |
| | 142 ROV PC33 | DEA west plough ridge | 7°04.4094' | 88°27.8330' | 4139 |
| | 163 ROV PC83 | DEA east plough ridge | 7°04.4926' | 88°26.9333' | 4143 |
| | 232 ROV PC64 | DEA south plough ridge | 7°04.6890' | 88°27.4554' | 4156 |





| | *DEA subsurface patch* | | | |
|---|---|---|---|---|
| 142 ROV PC48 | DEA west subsurface patch | 7°04.4113' | 88°27.8127' | 4140 |
| 169 ROV PC83 | DEA east subsurface patch | 7°04.4808' | 88°26.9130' | 4144 |
| 219 ROV PC58 | DEA south subsurface patch | 7°04.6930' | 88°27.4540' | 4155 |
| | *EBS track* | | | |
| 202 ROV PC80 | DEA west inside EBS track | 7°04.9533' | 88°28.1980' | 4150 |
| 202 ROV PC18 | DEA west side pile EBS track | 7°04.9609' | 88°28.1907' | 4150 |
| 211 ROV PC52 | DEA west rim inside EBS track | 7°04.9581' | 88°28.1909' | 4150 |
| 211 ROV PC73 | DEA west outside of EBS track | 7°04.9669' | 88°28.1929' | 4150 |
| **In situ oxygen profiles** | | | | |
| | *Reference* | | | |
| 158 LANDER-#1 | Reference South | 7°7.4590' | 88°26.9740' | 4155 |
| | *Undisturbed DEA* | | | |
| 176 ROV profiler #1 | DEA E outside track | 7°4.4677' | 88°26.9187' | 4102 |
| 169 ROV profiler #1 | DEA E outside track | 7° 4.4563' | 88°26.9176' | 4102 |
| 213 ROV profiler #2 | EBS outside track | 7°5.0220' | 88°28.1526' | 4189 |
| 146 ROV profiler #2 | DEA W outside track | 7°4.4000' | 88°27.8274' | 4140 |
| | *DEA plough tracks* | | | |
| 176 ROV profiler #2 | Subsurface | 7°4.4762' | 88°26.9190' | 4102 |
| 154 ROV profiler #2 | Subsurface | 7°4.4118' | 88°27.8172' | 4101 |
| 166 ROV profiler #2 | Furrow | 7°4.4898' | 88°26.9286' | 4104 |
| 169 ROV profiler #2 | Furrow | 7°4.4787' | 88°26.9205' | 4104 |
| | *EBS track* | | | |
| 202 ROV profiler #1 | Inside track | 7°4.9787' | 88°28.1730' | 4189 |





**Table 2**. Summary of analysed properties, analytical methods, estimated analytical errors, and detection limits

| Parameter | Method | Error (detection limit)[a] |
|---|---|---|
| $NO_3^-$ | Spectrophotometer (as sulphanile-a-naphthylamide)[b] | (1 µmol l$^{-1}$) |
| $NO_2^-$ | Spectrophotometer (as sulphanile-a-naphthylamide)[b] | (1 µmol l$^{-1}$) |
| $NH_4^+$ | Spectrophotometer (as indophenol blue)b | (1 µmol l$^{-1}$) |
| $Mn^{2+}$ | ICP-AES | 5-10% (1 µmol l$^{-1}$) |
| $SO_4^{2-}$ | Ion chromatography | 0.8-1.2 mmol l$^{-1}$ (5 mmol l$^{-1}$) |
| $C_{org}$ | CHN-Analyser[c] | 0.04 wt% |
| Alkalinity | Titration[d] | 0.05 meq l$^{-1}$ |
| Porosity | Weight difference before and after drying of the sediment | 0.02 |

[a] Note. For some properties no analytical error could be determined because of few data points or concentrations close to the detection limit.

[b] Grasshoff et al. (1999).

[c] Welicky et al. (1983).

[d] Breland and Byrne (1993).




**Table 3.** Reaction stoichiometry and rate expressions of the organic mater $((CH_2O)_a(NH_3)_b(H_3PO_4)_c)$ degradation and secondary redox reactions. All solute species are in concentrations of $\frac{mmol}{L_{pw}}$ and all solid species in $\frac{mmol}{L_{ds}}$. All reaction rate expressions are stated in the units of $\frac{mmol}{L_{pw} \, a}$ via conversion with $F_{pw} = \left(\frac{1-\phi}{\phi}\right)$.

| ID | Reaction | Rate expression (mmol/Lpw/a) |
|---|---|---|

***Organic matter degradation***

$(CH_2O)_a(NH_3)_b(H_3PO_4)_c + (a+2b)O_2$
$\qquad + (b+2c)HCO_3^- \rightarrow (a+b+2c)\,CO_2$
$\qquad + (b)NO_3^- + (c)HPO_4^{2-} + (a+2b+2c)H_2O$

$\qquad\qquad\qquad\qquad \sum_{i=1,2,3} k_i \, \frac{Corg_i}{a} \, R_{O2} \, F_{pw}$

$(CH_2O)_a(NH_3)_b(H_3PO_4)_c + \left(\frac{4}{5}a + \frac{3}{5}b\right)NO_3^-$

$\qquad \rightarrow \frac{1}{2}\left(\frac{4}{5}a + \frac{3}{5}b + b\right)N_2 + cHPO_4^{2-} + \left(\frac{3}{5}a + \frac{6}{5}b + 2c\right)H_2O$

$\qquad + \left(\frac{1}{5}a - \frac{3}{5}b + 2c\right)CO_2 + \left(\frac{4}{5}a + \frac{3}{5}b - 2c\right)HCO_3^-$

$\qquad\qquad\qquad\qquad \sum_{i=1,2,3} k_i \, \frac{Corg_i}{a} \, R_{NO3} \, F_{pw}$

$(CH_2O)_a(NH_3)_b(H_3PO_4)_c + 2a\,MnO_2 + (3a+b-2c)CO_2$
$\qquad + (a+b-2c)H_2O$
$\qquad \rightarrow (4a+b-2c)HCO_3^- + 2a\,Mn^{2+} + bNH_4^+ + cHPO_4^{2-}$

$\qquad\qquad\qquad\qquad \sum_{i=1,2,3} k_i \, \frac{Corg_i}{a} \, R_{MnO2} \, F_{pw}$

***Secondary redox reactions***

$NH_4^+ + 2O_2 + 2HCO_3^- \rightarrow NO_3^- + 2CO_2 + 3H_2O$ $\qquad\qquad k_{NH4Ox} \, NH_4 \, O_2$

$2Mn^{2+} + O_2 + 4HCO_3^- \rightarrow 2MnO_2 + 4CO_2 + 2H_2O$ $\qquad\qquad k_{MnOx} \, Mn^{2+} \, O_2$

$4Fe(II) + O_2 + 2H_2O + 4CO_2 \rightarrow 4Fe(III) + 4HCO_3^-$ $\qquad\qquad k_{Fe(II)Ox} \, Fe^{2+} \, O_2$

***Monod expressions***

$R_{O2} = \dfrac{O_2}{K_{O2} + O_2}$

$R_{NO3} = \dfrac{NO_3}{K_{NO3} + NO_3} \dfrac{K_{o2}}{K_{O2} + O_2}$

$R_{MnO2} = \dfrac{MnO_2}{K_{Mn} + MnO_2} \dfrac{K_{NO3}}{K_{NO3} + NO_3} \dfrac{K_{o2}}{K_{O2} + O_2}$




**Table 4**. Constrained and fitted parameters used in the numerical simulations

| Parameter | Value | Units | Reference |
|---|---|---|---|
| ***General parameter*** | | | |
| Temperature | 4 | °C | a) |
| Pressure | 400 | bar | a) |
| Salinity | 35 | ‰ | a) |
| Redflied ratio (C:N:P) | 106:16:01 | | |
| Sedimentation rate | 0.0004 | cm a$^{-1}$ | Haeckel et al. (2001) |
| Maximu depth of calculation | 200 | cm | |
| Number of points in the numerical grid (uneven) | 500 | | |
| ***Fitted parameters - background (reference) model*** | | | |
| Porosity at sediment surface ($\varphi_0$) | 0.94 | | c) |
| Porosity at infinite depth ($\varphi_\infty$) | 0.86 | | c) |
| Porosity attenuation coefficient (β) | 0.14 | | c) |
| Bioturbation coefficient ($D_b^0$) | 0.65 | cm$^2$ a$^{-1}$ | b) |
| Bioturbation half depth ($x_{Db}$) | 10 | cm | b) |
| Bioturbation decrease ($\beta_{Db}$) | 4 | | b) |
| Flux of labile $C_{org}$ ($F_{G0}$) | 10 | µmol cm$^{-2}$ a$^{-1}$ | c) |
| Flux of semi-labile $C_{org}$ ($F_{G1}$) | 1.2 | µmol cm$^{-2}$ a$^{-1}$ | c) |
| Flux of refractory $C_{org}$ ($F_{G2}$) | 0.06 | µmol cm$^{-2}$ a$^{-1}$ | c) |
| Rate constant for labile $C_{org}$ oxidation ($k_{G0}$) | 0.1 | a$^{-1}$ | c) |
| Rate constant for semi-labile $C_{org}$ oxidation ($k_{G1}$) | 0.003 | a$^{-1}$ | c) |
| Rate constant for refractory $C_{org}$ oxidation ($k_{G2}$) | 5E-07 | a$^{-1}$ | c) |
| Rate constant for reaction of $Mn^{2+}$ and $NO_3^-$ | 0.2 | l mmol$^{-1}$ a$^{-1}$ | c) |
| Rate constant for reaction of $Mn^{2+}$ and $O_2$ | 1E+06 | l mmol$^{-1}$ a$^{-1}$ | c) |
| Rate constant for reaction of $NH_4^+$ and $O_2$ | 500 | l mmol$^{-1}$ a$^{-1}$ | c) |
| Monod constant for $O_2$ reduction | 8E-06 | mmol | c) |
| Monod constant for $NO_3^-$ reduction | 0.03 | mmol | Boudreau (1996) |
| Monod constant for $MnO_2$ reduction | 10 | mmol | Boudreau (1996) |
| Monod constant for $SO_4^{2-}$ reduction | 1 | mmol | Boudreau (1996) |
| [$MnO_2$]$_{model\ domain}$ | 1.9 | wt% | a) |
| [$O_2$]$_{bottom\ water}$ | 0.132 | mmol l$^{-1}$ | a) |
| [$NO_3^-$]$_{bottom\ water}$ | 0.042 | mmol l$^{-1}$ | a) |
| [$NH_4^+$]$_{bottom\ water}$ | 0 | mmol l$^{-1}$ | a) |
| [$Mn^{2+}$]$_{bottom\ water}$ | 0 | mmol l$^{-1}$ | a) |
| [Alkallinity]$_{bottom\ water}$ | 2.4 | meq l$^{-1}$ | a) |
| [$Mn^{2+}$]$_{lower\ boundary}$ | 0.05 | mmol l$^{-1}$ | a) |
| ***Parameter for impact simulations*** | | | |
| Bioturbation recovery time ($tD_B$) | 100 / 200 | a | d) |
| *Case 'reactive Fe(II)-O2 layer'* | | | |
| [Fe(II)] at lower boundary | 0.44 | wt% | e) |
| Fe(II) half depth of increase | 20 | cm | |
| Fe(II) profile reduction coefficient | 2 | | |
| *Case 'removal'* | | | |
| Removed sediment thickness | 10 | cm | d) |
| *Case 'O2 flooding'* | | | |





| | [O2]bottom water | mmol l⁻¹ | |
|---|---|---|---|
| $[O2]_{reactive\ layer}$ | | | |
| Half depth O$_2$ decrease | 12 | cm | |
| O$_2$ attenuation coefficient | 2.5 | | |
| ***Bioturbation model*** | | | |
| | | | |
| $^{210}$Pb radioactive decay constant | ln(2)/22.3 | a⁻¹ | |
| $^{226}$RaTh radioactive decay constant | ln(2)/1602 | a⁻¹ | |
| $^{230}$Th radioactive decay constant | ln(2)/57380 | a⁻¹ | |
| $[^{210}$Pb$]_{bottom\ water}$ | 1.0 | Bq g⁻¹ | b) |
| $[^{226}$Ra$]_{bottom\ water}$ | 0.3 | Bq g⁻¹ | b) |
| $[^{230}$Th$]_{bottom\ water}$ | 0.6 | Bq g⁻¹ | b) |

a) Approximated by field data; b) fitted to Pb data (Figure 7); c) fitted to field data; d) Approximated from data and references presented in Volz et al. (submitted); e) Inferred from data presented in König et al. (1997, 1999)





**600**     **Figures**

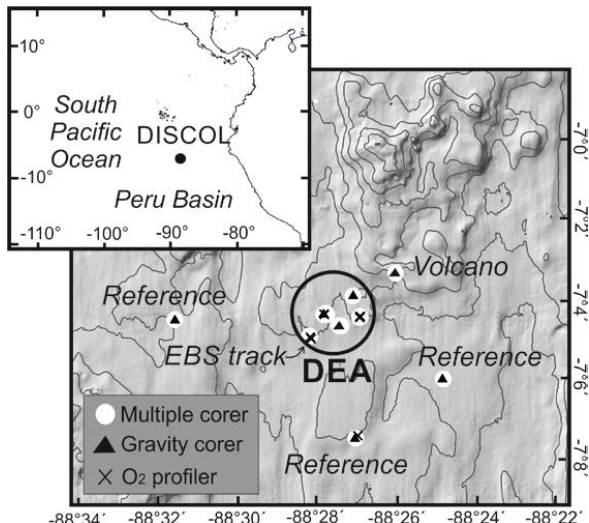

**Figure 1.** Bathymetric map of the DEA region in the Peru basin including the sampling stations presented in this

work. More details on the sampling locations are provided in Table 1.






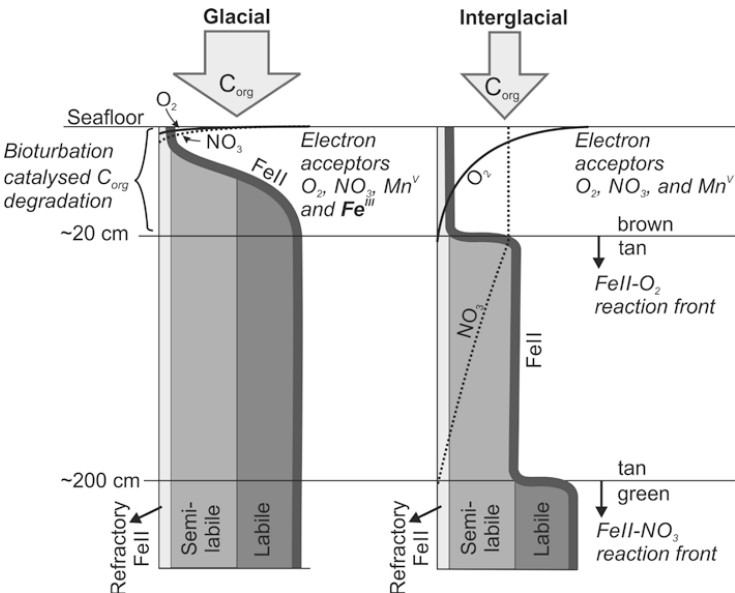

**Figure 2**. Schematic diagram (not to scale) outlining the formation of a Fe(II) rich clay phase (green) in the context of glacially high organic matter input and its conversion to Fe(III) by downward progressing $NO_3$ fronts

during interglacial periods triggered by reduced organic matter input as postulated by König et al. (2001). In this study, we propose that a second reaction front exists impeding the downward $O_2$ migration through a reaction with the semi-labile Fe(II) phase.








**Figure 3**. Informative solute profiles, organic carbon content and porosity in gravity cores and their corresponding multicores. Symbols represent the measured values at the mean depth of the sediment layer sampled. Details on individual sampling stations can be found in Table 1.



**Figure 4**. Solute profiles in multi- and box-corer retrieved in various microhabitats. Symbols represent the measured values at the mean depth of the sediment layer sampled. Details on individual sampling stations can be found in Table 1.






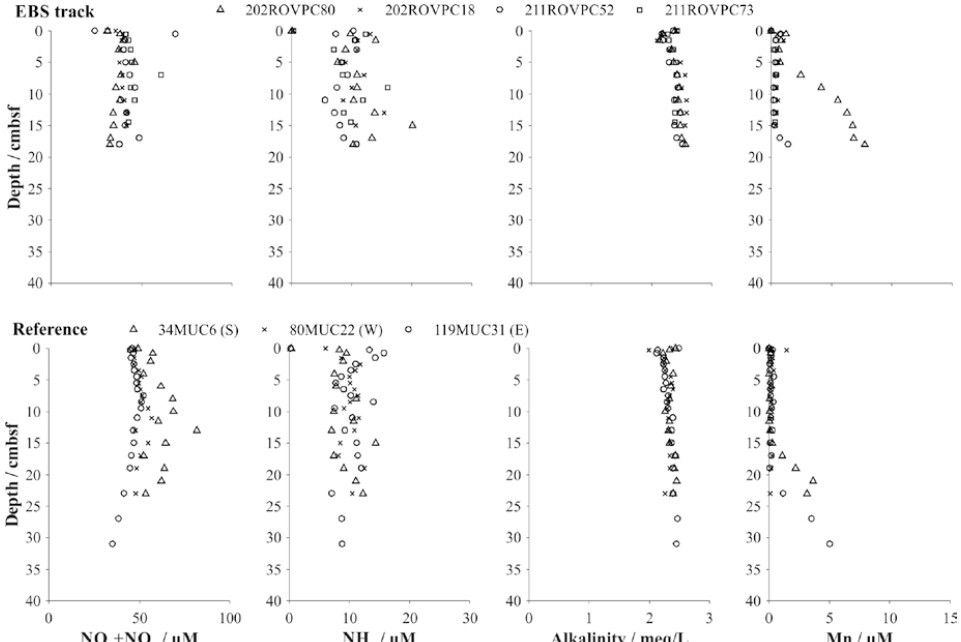

**Figure 4**. Cont'd.

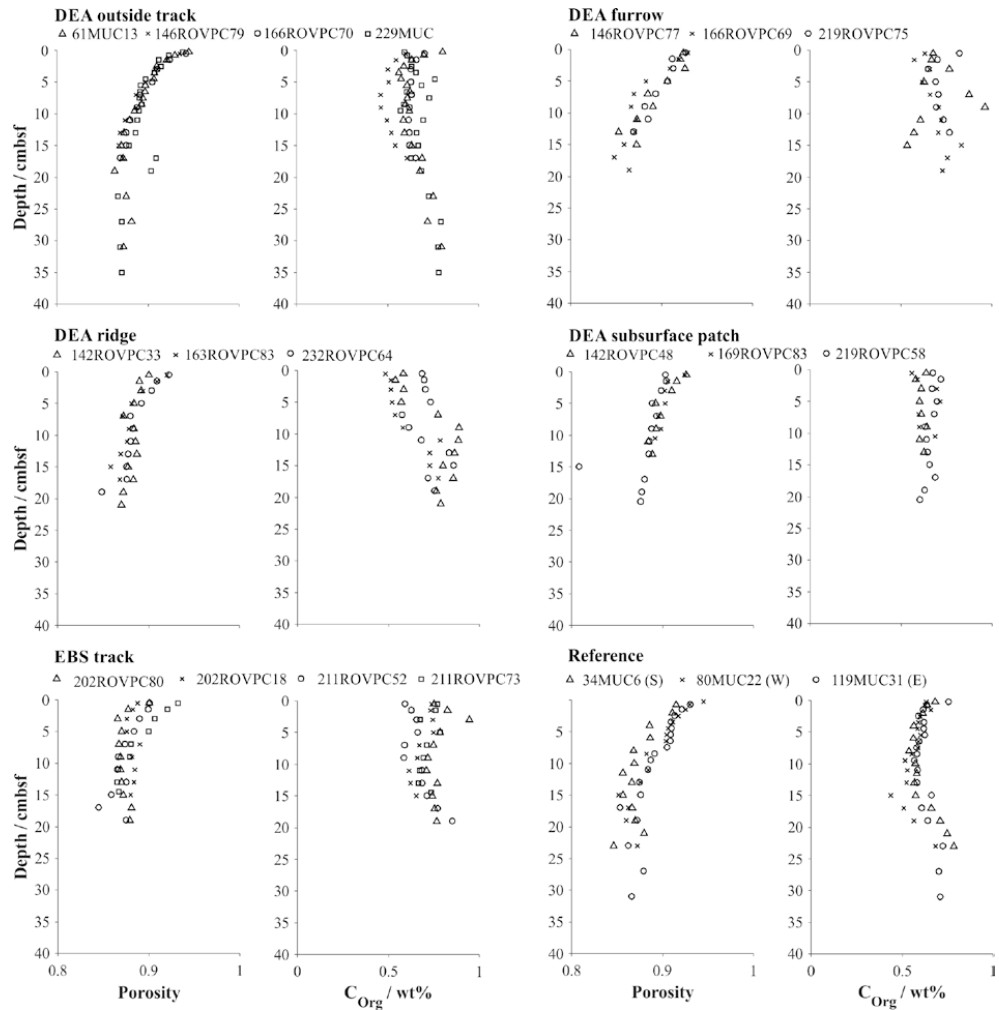

**Figure 5**. Porosity and organic carbon content in multi- and box-corer retrieved in various microhabitats. Symbols represent the measured values at the mean depth of the sediment layer sampled. Details on individual sampling stations can be found in Table 1.





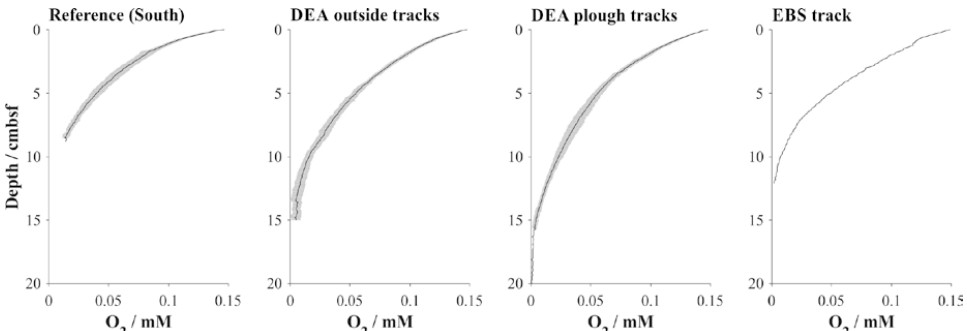


**Figure 6**. Averaged (black line) in situ oxygen profile representative of various disturbance settings including their standard deviations (shaded area). Because the depth steps between each measuring point of the oxygen profiles are very close, they are not resolved in the diagram. Details on individual sampling stations can be found in Table 1.






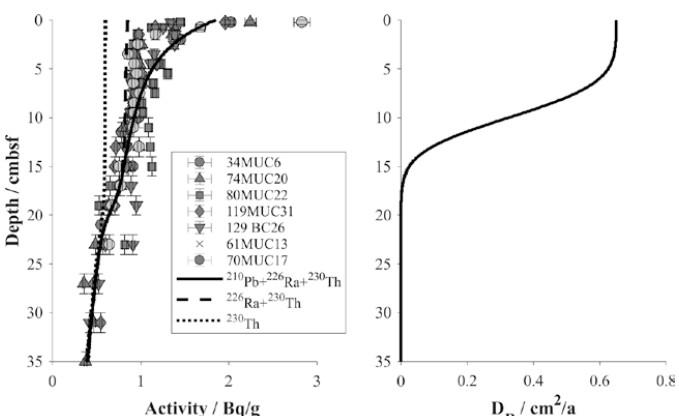

**Figure 7**. Model results of the faunal bioturbation activity in reference and DEA-outside track sediments based on measured $^{210}$Pb activities. Left: Model curves indicate total $^{210}$Pb activity (solid) and supported contributions by $^{230}$Th (dotted) and $^{226}$Ra (dashed minus dotted). Right: Bioturbation intensity profile derived for the background model.






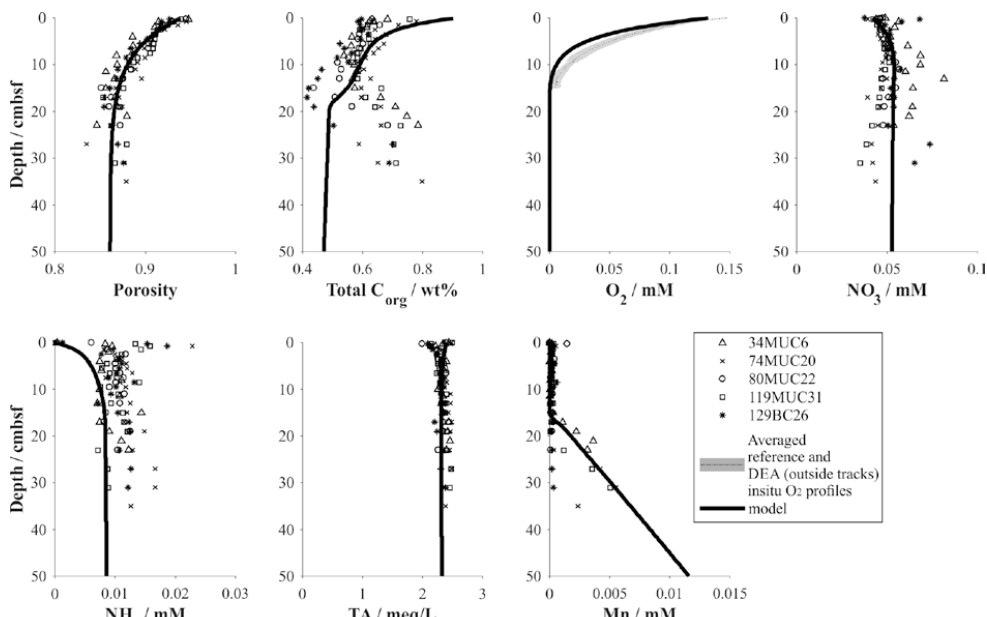

**Figure 8**. Simulation of background biogeochemical processes and their effect on solute profiles in the DEA
region. See Table 4 for parameterization of the modelled profiles.






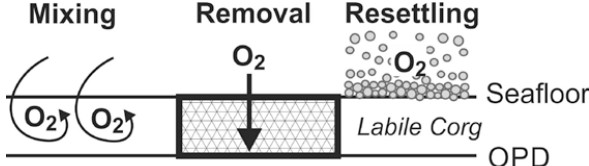

**Figure 9**. Schematic characterization of the different impact types at a mined seafloor and their influence on $O_2$ distribution.


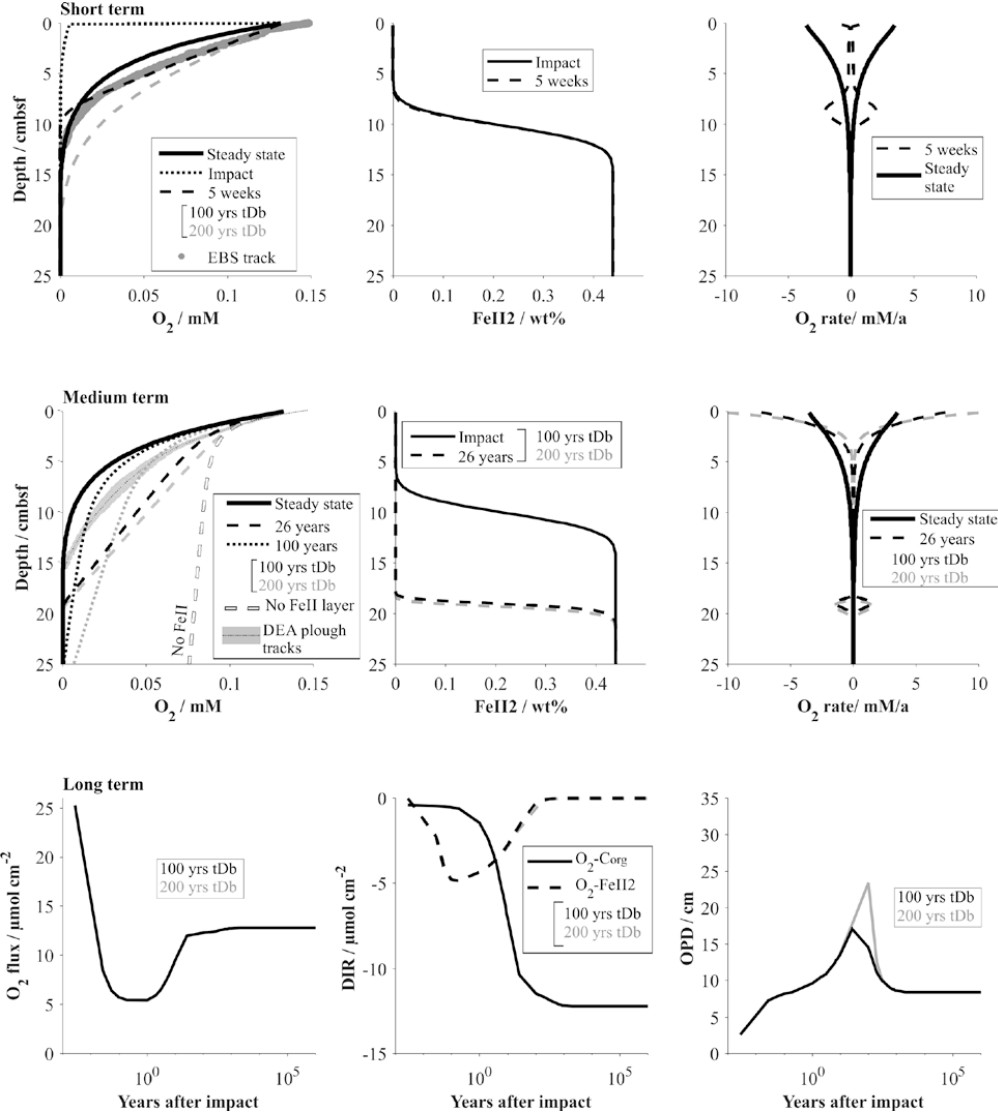

**Figure 10**. Impact simulations after a disturbance causing the reactive upper 10 cm to be removed. The short term (up to 5 weeks, comparable to the EBS tracks) and medium term (26 years, comparable to the DEA tracks) impact is visualized by the respective $O_2$ profiles and the fluxes shaping the $O_2$ profiles. It is assumed that the oxygen penetration depth is impeded by a reactive Fe(II) layer. Note, the Fe(II) is assumed to be located at 20 cmbsf but is placed at 10 cm if the upper 10 cm are removed. For comparison the $O_2$ profile in the absence of a Fe(II) layer is also shown. The long term impact is demonstrated by the surface flux of oxygen into the sediment, the depth integrated reaction rates (DIR) of oxygen with organic matter ($C_{org}$) and with the reactive Fe(II) layer, as well as the oxygen penetration depth (OPD) over time. The influence of the bioturbation recovery time is shown by plotting the lines that correspond to a recovery interval of 100 and 200 years in black and grey respectively.



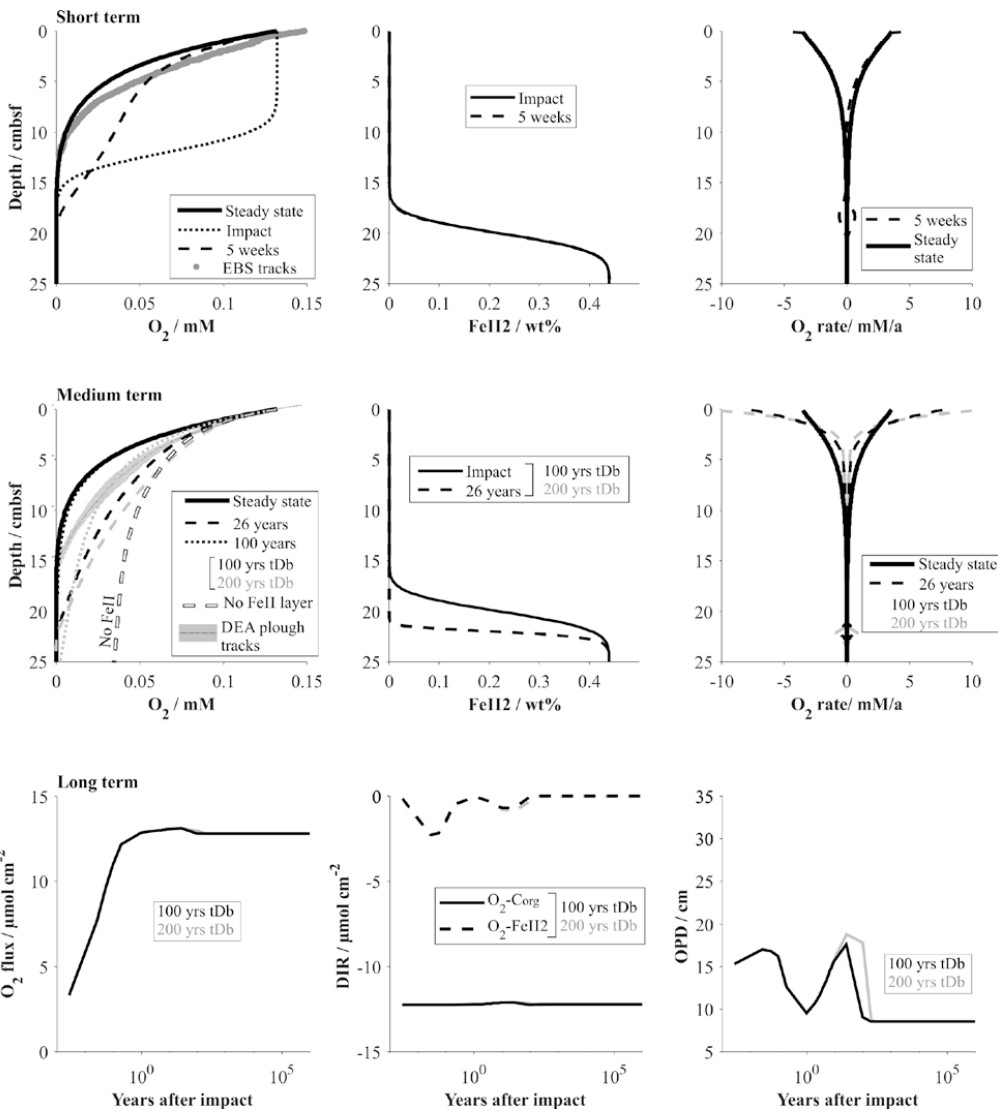

**Figure 11**. Impact simulations after a disturbance causing the upper sediment layer to be exposed to bottom water oxygen levels (i.e. through mixing or by resettling sediments, see Figure 9). See the caption of Figure 10 for details.





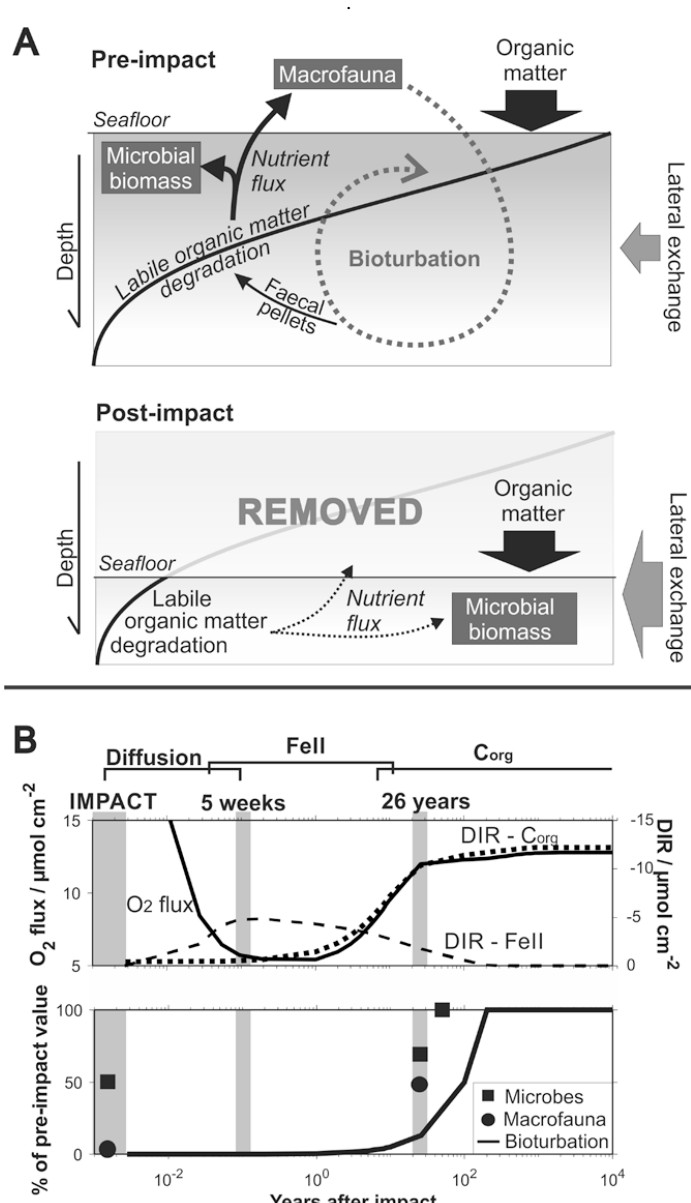

**Figure 12**. (A) Schematic diagram showing the delicate interplay between the distribution of organic matter, microbial activity and the bioturbation through burrowing megafauna in a pre-impact system and after removal of the upper reactive sediment layer. (B) Quantitative comparison of geochemical and biological processes over time after an impact through sediment removal. Biological data is taken from Stratmann et al. (2018) (Macrofaunal abundance) and Vonnahme et al. (submitted) (microbial cell cout) for the most severely disturbed sites (e.g. inside 5 weeks old EBS tracks, and inside 26 years old plough marks).