# Peer review of "DISCOL experiment revisited: Assessing the temporal scale of deep-sea mining impacts on sediment biogeochemistry"

_Biogeosciences, 2019_

## Referee Comment (RC1) · Anonymous Referee #1 · 21 Oct 2019

This paper covers an important subject, which should be of interest to the readers of Biogeosciences. The major finding of this study is the differences in recovery time of deep sea sediments between two methods that can be used during deep sea mining. The authors used an extensive dataset and model porewater and solid phase depth profiles with numerical a numerical model.

In general I think the structure of the paper is a bit confusing. There is an introduction section (section 1) but within this section there is a long discussion/introduction of the general biogeochemistry of the Peru Basin (section 1.1). I think this would be more in place in the methods section as a site description. The results section contains a lot of

discussion where I think it should be better to first clearly describe all the data, which is used in this study. The discussion section is rather short and reads as a conclusion part. I think the discussion and conclusion part can be combined. I think the manuscript is highly under referenced.

Abstract: Line 19: "reduces seafloor nutrient fluxes over centuries" - I think based on this study the authors can only draw conclusions for oxygen. Otherwise the authors should include more data from their modelling study on the effects of these two different sampling techniques. Now the only information given in the discussion (and figures 10 and 11) are oxygen fluxes. I think it would be a good addition to also show other nutrients fluxes from the sediment (for example ammonium/nitrate), which are already in the model I guess. - Can the authors also give some information about the changes/recovery of macrofaunal communities.

Introduction In my opinion section 1.1 is a bit out of place. This section describes processes that are important in general (degradation of organic matter P2 lines 55-59). I would move this to the introduction and make a new section 2.1 with a site description where you discuss all the processes that are important for your study. Also I think this part is under referenced.

In lines 39-41 you describe there were studies that focused on characterization and distribution of macrofauna. I think it would good to shortly explain what was found in these studies in this section because disturbance of macrofauna is important as you also describe in you abstract. I think the last paragraph of this section is a bit out of place or should be better introduced.

Results: It is quite difficult to understand to which plots you refer in this section. Add more figure numbers behind your statements. Also the order of discussing data is a bit strange. You start with porosity but this is plotted in figure 5. You should start with describing figure 3 or change the order of figures.

There is a lot of discussion in section 3.1. I think it would be better to first describe

the porewater and solid phase profiles (in order of the figures) before you start your discussion. Section 3.3 Most of the interesting statements in this section are based on Vonnahme et al submitted or at visual observations. There is not a good link with the data shown in figures 4-6. I think it would be better to compare these instead of discussing visual observations or "repeating" findings presented in Vonnahme et al. submitted.

Conclusions I suggest to strongly shorten this section and only discuss the main conclusions found in this study and not discuss conclusions from other studies.

Figures: The resolution of the figures could be improved.

Detailed comments: Line 17: "(micro)-biological data". Is this in place? There is no microbiological data in this manuscript.

Line 18: how was the depth of this reactive surface layer quantified.

Line 19: "reduces seafloor nutrient fluxes over centuries". From sediment towards the water column? Maybe say benthic release of nutrients.

Line 39: col in recolonization should be in capitals similar to dis in disturbance

Line 43: introduce abbreviation of ROV

Line 47: What do the authors mean with a "state of the art geochemical dataset". I think it would be more appropriate to call the data set extensive, which is indeed the case when taking into account the amount of profiles. However, I would suggest to also include porewater iron as this variable is discussed multiple times throughout the manuscript.

Line 61-63: how deep is the layer where there is organic matter degradation? Does this mean there is no iron of sulfate reduction?

Line 64: which depth is the bioturbation depth approximately?

Line 65: "Oxygen is depleted at a sediment depth of 5 to 15 cm, " Do you mean in general for deep see sediments? Then add a reference.

Line 65: "with the DEA region being on the deeper end". Add reference.

Line 66: "In the slightly shallower region of the Peru Basin seawater-derived nitrate and nitrate from oxic respiration is consumed within the bioturbated upper 20 cm of sediment". Add reference.

Line 70: It is a bit confusing to also refer to this layer as reaction layer as you also have a surface 'reaction layer'. I would use a different name iron-rich layer for example.

Line 70: "formed during glacial times". Approximately how long ago?

Line 70: add reference to :" This reaction layer was formed during glacial times, when organic matter deposition was strongly increased and potential electron acceptors such as oxygen, nitrate and manganese oxide were exhausted closely below the sediment surface. "

Line 79-85: Is this one of the main research questions? This is not described in the abstract.

Line 89: "to artificially disturb the". This means without removing the reactive surface layer?

Line 92: what does less disturbed mean. The sediment depth of disturbance?

Line 97: I would give this information directly when you describe "artificially disturbance".

Line 115: 20 mictoliter of HCl was added to how much sample? 1 mL?

Line 116: "Additional sediment samples ". Do you mean a separate sediment core or the sediment leftovers after porewater extarction?

Line 119: you also measured show manganese data so you should add this to the list.

Line 119-124: Can you add the sample sizes to this paragraph or add this in Table 2.

Line 154: Can you add equation numbers to all equations? That makes it easier to refer to them in the text.

Line 176: shortly describe how the gridsize is distributed.

Line 178: describe that you use organic matter with different reactivities (Multi G approach) and give references for this approach (Jørgensen, 1978;Westrich and Berner, 1984; Middelburg, 1989).

Line 195: I do not see a sharp drop in porosity in figure 5.

Line 195: You should also give the reference to the figure.

Line 195: "and is rich in manganese". Can you add a concentration to this statement, or a range of concentrations.

Line 200: "Organic carbon content in the DEA region oscillates about a mean value of 0.5 to 0.75 wt% in the upper 50 cm,". refer to figure.

Line 202: "With depth the organic carbon content decreases steadily to about 1 wt% at 10 m.b". Add figure reference. Maybe I'm not looking at the right figure but the organic carbon content in fig 3 decreases to almost 0 and is never above 1 wt%.

Line 203-207: This part seems more suitable to the discussion. I think it is better to first discuss the data you show in figures 3-6.

Line 208-210: This is discussion, focus on presenting the data first.

Line 217: "Dissolved manganese is strongly redox sensitive and is absent in the upper oxic zone." Add reference to such a statement.

Line 226-228: Where can I see this Fe data? If you don't want to add this to the manuscript you can add it in an appendix.

Lines 231-235: I would add this part to the methods.
Line 238: There should be a dot after inhomogeneities, Not a comma.

Line 241: Can you further explain why you choose these sites?

Line 248: "which suggests that secondary reactions, such as a semi-labile Fe(II)-oxygen reaction front, additionally shape the oxygen profile:" Why is this reaction not added in the model?

Line 260: "relevant information" is a bit vague.

Line 264-266: Is this data compared to the reference stations?

Line 268: "signs of bioturbation are lacking". how would you see these and are these sign visible at the reference site?

Line 278: Can you give macrofaunal density (add a number).

Line 292-295: These lines are repetition. This is already discussed in the introduction.

Lines 300-304: For me it is difficult to understand how these changes were modeled. Can this be explained in the method section in more detail?

Line 305: "It is assumed that bioturbation is inhibited immediately after the impact with a linear increase to undisturbed reference bioturbation intensity within 100 to 200 years". Where is this assumption based on, can you give a reference?

Line 311: "Upper centimetres". Give a number.

Line 321: For me it is quite surprising that there is such a large difference in such a short period.

Line 346: "overall surface fluxes vary". Where can I see the all the fluxes besides oxygen?

Line 367 (Discussion): first paragraph does not give any new information or does discuss the results found in this study. The discussion reads as a summary/conclusion section.

Line 370: "The associated microbial growth". Add references

Line 379: "..infauna new material for additional degradation and microbial colonization, close the causal cycle." Add reference

Lines 386-388:" through changing the physical characteristics of the sediment surface, i.e. by exposing more compacted sediments with increased shear strength, and removing manganese nodules as hard substrate habitats." Add reference.

Line 389: Where can I find the biological data?

Table 4: - "Number of points in the numerical grid (uneven)". How are they distributed? - "Rate constant for reaction of Mn2+ and NO3". This equation is not given in table 3 - How do the rate constants (last column) compare with values used in the literature? Are they in the same range? I think it would be good to add a column which described the ranges of rate constants used in the literature (with references). - there is no rate constant describing the third secondary reaction (oxidation of Fe2+) - "Monod constant for SO42- reduction". This reaction is not in table 3. If you add this reaction you should also add Fe reduction coupled to organic matter.

Figure 8: - Total Co org. Not a very good fit. You could add more refractory OM deposition during the glacial periods. Also I would like to see how the three fractions are distributed with depth - For me it is unclear why/how the selection of stations is made to fit the model. The stations are both inside and outside the DEA region.

Figure 10 - FeII2: Why not show the steady state? I think it would be better to use the same type of line for impact in all the plots.
* * *

---

## Referee Comment (RC2) · Anonymous Referee #2 · 31 Dec 2019

Haffert and co-authors present a comprehensive sedimentary geochemical dataset on an experimentally disturbed potential deep-sea mining area, called DISCOL. The paper reports an extensive set of downcore geochemical data (O2, organic C, nutrients) from both short (MUC, box corer) as well as long (GC) cores in the experimental site. Further, the study improves an existing diagenetic reaction-transport model with data-driven process optimisations. The authors then use these new data plus transient early diagenetic simulations to asses the short and long-term impact of sediment removal during (future) mining of polymetallic nodules. They find that the removal of the surface labile organic carbon along with the nodules is the single most important driver of the establishment of a new geochemical regime in the disturbed areas.

[Figure]

The primary strength of this work is a quite convincing set of predictions as to what will happen in the long run when these nodules will be extracted from the seabed. The transient simulations are very useful and well integrated to the data. Supported by an extensive and novel dataset and the improved modelling approach, the integrated methodology could be used in other seafloor resource extraction scenarios as well.

There is no significant weakness in the manuscript. It is well laid out and well written. I only have a few suggestions for a minor revision of the existing manuscript:

Overall comment: How does the Fe(II)-rich clay layer can trap nitrate? The redox reaction between the two is not that well established, and wondering if a more complex cycle is present here, involving nitrogen intermediate species and a more complicated Fe(II)-Fe(III) cycle. I would propose that Figure 2 can be improved to clarify this, and both introduction (L84) and discussion parts can be expanded with a more detailed proposition of redox pathways.

L27: JPI Oceans - with plural 'oceans', I think is the right acronym.

L54: the work 'untypically' can be removed without a significant change in the meaning of the sentence.

L58: 'availability positions' - perhaps could be re-phrased using 'order of the electron acceptors' or similar.

Section 1.1 - overall I find this section is more like a discussion, rather than introduction. As mentioned above, the introduction of a Fe(II)-NO3 redox pathway is a little bit out of place in this section. Besides, the readers might expect a following section of 1.2, since there is 1.1, bot there is no other subsection of the introduction. Please consider re-organizing the material in 1.1.

L200-205 - interesting - did all cores include such buried nodule layers? I would strongly recommend to indicate the depth of these layers in the Figures 3-6 to be able to see directly in the figure if the nodule is impacting the geochemical profiles.

---

## Author Comment (AC1) · 21 Jan 2020

Anonymous Referee #1 This paper covers an important subject, which should be of interest to the readers of Biogeosciences. The major finding of this study is the differences in recovery time of deep sea sediments between two methods that can be used during deep sea mining. The authors used an extensive dataset and model porewater and solid phase depth profiles with numerical a numerical model.

In general I think the structure of the paper is a bit confusing. There is an introduction section (section 1) but within this section there is a long discussion/introduction of the general biogeochemistry of the Peru Basin (section 1.1). I think this would be more in

place in the methods section as a site description. The results section contains a lot of discussion where I think it should be better to first clearly describe all the data, which is used in this study. The discussion section is rather short and reads as a conclusion part. I think the discussion and conclusion part can be combined. I think the manuscript is highly under referenced.

- As described in more detail in later responses to comments directly related to each section, we will restructure the document and suitable references will be added.

Abstract: Line 19: "reduces seafloor nutrient fluxes over centuries" - I think based on this study the authors can only draw conclusions for oxygen. Otherwise the authors should include more data from their modelling study on the effects of these two different sampling techniques. Now the only information given in the discussion (and figures 10 and 11) are oxygen fluxes. I think it would be a good addition to also show other nutrients fluxes from the sediment (for example ammonium/nitrate), which are already in the model I guess.

- Figure 10 and 11 focus on oxygen profiles because of the central role that the oxygen penetration depth plays. Figure 12 depicts seafloor fluxes of nutrients and here we will add the seafloor fluxes of NH4 and NO3 over time to the flux diagram 12B.

Can the authors also give some information about the changes/recovery of macrofaunal communities.

- We will add a sentence at the end of the abstract addressing faunal recovery: 'The interdisciplinary (geochemical, numerical and biological) approach highlights the closely linked nature of benthic ecosystem functions, e.g. through bioturbation, microbial biomass and nutrient fluxes, which is also of great importance for the system recovery. It is, however, important to note that the nodule ecosystem may never recover to pre-impact state without the essential hard substrate and will instead be dominated by different faunal communities, functions and services'. More specific analyses of changes / recovery of macrofauna is beyond the scope of this study. Introduction:

In my opinion section 1.1 is a bit out of place. This section describes processes that are important in general (degradation of organic matter P2 lines 55-59). I would move this to the introduction and make a new section 2.1 with a site description where you discuss all the processes that are important for your study. Also I think this part is under referenced.

- We will move this section to the methodology section and rename it to 'Biogeochemical site description'. Only very few studies give detailed information on the biogeochemistry of the area. The repetitive use of these reference throughout the text is impracticable so that the main references are introduced before the specific site description: 'König et al. (1999), König et al. (2001) and Haeckel et al. (2001) have previously identified and quantified through diagenetic modelling the main biogeochemical processes in the Peru Basin, including the DEA region.' We will add a colon after this sentence to emphasize that the following information belongs to these references.

In lines 39-41 you describe there were studies that focused on characterization and distribution of macrofauna. I think it would good to shortly explain what was found in these studies in this section because disturbance of macrofauna is important as you also describe in you abstract.

- We will add references to the sentence (Line 41): The DISCOL experimental area (DEA) was revisited before and several times after the initial disturbance, with research concentrating mainly on the characterization and distribution of benthic fauna (e.g. Borowski and Thiel, 1998; Borowski, 2001; Bluhm, 2001). And we will add here a sentence on macrofaunal abundances: 'Macrofaunal communities, which are particularly important for the bioturbation and thus redox zoning of the sediment, are mainly composed of Polychaeta (about 50%) and to a lesser extent tanaidacea (10 - 20%) and bivalvia (5 – 10%) (Borowski and Thiel, 1998). After the artificial disturbance, very few faunal groups returned to baseline or control conditions after more than two decades (Jones et al., 2017) and the main mode of macrofaunal recolonization was found to be via lateral migration (Borowski and Thiel, 1998).'

I think the last paragraph of this section is a bit out of place or should be better introduced.

- We will move this section, which intends to justify the introduction of a shallow Fe(II)-O2 reaction layer, to the description of the diagenetic model to Line 180: 'The reaction network shaping the geochemistry of the upper sediment metres, namely organic matter degradation and secondary redox reactions including the oxidation of ammonium and dissolved manganese, are listed in Table 3 as well as the corresponding rate expressions. We also included a shallow semi-labile Fe(II) layer that reacts with oxygen to the model (Figure 2). While it is established that a labile Fe(II) reaction front impedes the downward migration of nitrate (König et al., 1997;König et al., 1999), little is known about the redox potential of Fe(II) on different crystallographic lattice positions (coordination sites) in clay minerals, such as nontronite. In particular, information is lacking on the reactivity of the Fe(II) on these sites with respect to oxidation by nitrate and oxygen. Available Mößbauer data (König et al., 2001) from the DISCOL area indicate a refractory fraction of Fe(II) that prevails in the nitrate reduction zone ($\sim$10 % of total iron). In the light of the much higher redox potential of oxygen compared to nitrate, we model the remaining Fe(II) as a semi-labile phase representing a second reaction front impeding the downward migration of oxygen.'

Results: It is quite difficult to understand to which plots you refer in this section. Add more figure numbers behind your statements.

- We will add more Figure references to the statements.

Also the order of discussing data is a bit strange. You start with porosity but this is plotted in figure 5. You should start with describing figure 3 or change the order of figures.

- We will reorder Figure 3-5, beginning with the solid composition of the sediment.

There is a lot of discussion in section 3.1. I think it would be better to first describe

the porewater and solid phase profiles (in order of the figures) before you start your discussion.

- In section 3.1 observed porewater trends are described in the context of well-established processes (e.g. nitrate decline through denitrification in anoxic sediments; relating manganese oxide reduction to the increase in dissolved manganese in anoxic sediments; linear profile shape indicates reaction front, etc). The structure of the results follows other examples in this field (e.g. Haeckel et al., 2001) and we believe that by providing a process-oriented description of the results, the discussion can better focus on the actual objectives of this manuscript.

Section 3.3 Most of the interesting statements in this section are based on Vonnahme et al submitted or at visual observations. There is not a good link with the data shown in figures 4-6. I think it would be better to compare these instead of discussing visual observations or "repeating" findings presented in Vonnahme et al. submitted.

- We would like to disagree with the reviewer in this point. Section 3.3 is a chapter on impact characterization. It includes (i) a physical description of the disturbance, (ii) references to result figures, where relevant differences of the impacted to reference sites can be noticed (oxygen profiles, porosity profiles, dissolved manganese) and (iii) information on a parallel microbiological study (Vonnahme et al, submitted), which is by nature closely interlinked with the biogeochemical processes discussed in this manuscript. Information on microbiological activities (the Vonnahme et al. paper) are not 'repeated' in a separate paragraph but are instead added throughout the text supporting the characterization of different impact types.

Conclusions I suggest to strongly shorten this section and only discuss the main conclusions found in this study and not discuss conclusions from other studies.

- The conclusion paragraphs are structured as follows: i.Highlighting the importance of the updated diagenetic transport-reaction model, especially with respect to the newly introduced Fe(II)-O2 reaction layer. ii.Summarizing the outcome of the prognostic

model by simply listing the three stages of recovery iii. A holistic (interdisciplinary) summary of the mining impact, highlighting the close link between geochemical and biological processes These three paragraphs directly link to the three research aims in the introduction. The last paragraph of the conclusion draws the larger picture of the impact of nodule mining in the deep-sea, which is not only applicable beyond the studied site but is also important for the non-expert readers who base regulatory decisions on publications like these. The authors have carefully structured the conclusions, which are in line with the guidelines of the publishers. We would thus like to maintain the structures as it is.

Figures: The resolution of the figures could be improved.

- The resolution issue is an artefact of the pdf converter of the journal webpage. The original files have much higher resolution.

Detailed comments: Line 17: "(micro)-biological data". Is this in place? There is no microbiological data in this manuscript.

- Microbiological abundances are stated throughout the text (for example in the 'impact characterisation') and are related to the availability of labile organic matter and consequences for the food web structure in the discussion. Figure 12 also plots some microbial abundances for comparison.

Line 18: how was the depth of this reactive surface layer quantified.

- The reactive layer was determined, on one hand, simply by its distinct dark brown colour and, on the other hand, by the oxygen penetration depth. This is described in the section 'Geochemistry of the DEA region: 'The upper reactive sediment section (up to ~20 cm.b.s.f) is markedly different from the deeper sediment sections. It is characterized by a sharp decrease in porosity (from 0.94 to approx. 0.86, Table 4) and is rich in manganese oxides (Paul et al., under review), which gives this layer its distinct dark brown color.[. . .] In situ oxygen profiles confirm that the upper brown layer

represents the oxygenated zone (Figure 6). In this zone, oxic respiration of organic matter not only consumes downward diffusing oxygen [. . .]'.

Line 19: "reduces seafloor nutrient fluxes over centuries". From sediment towards the water column? Maybe say benthic release of nutrients.

- We will change the sentence accordingly

Line 39: col in recolonization should be in capitals similar to dis in disturbance

- Yes indeed, the 'col' should have been capitalized which will be amended.

Line 43: introduce abbreviation of ROV

- We will change it to '[. . .] remotely operated vehicle (ROV) [. . .]'

Line 47: What do the authors mean with a "state of the art geochemical dataset". I think it would be more appropriate to call the data set extensive, which is indeed the case when taking into account the amount of profiles.

- For the research aim at hand, the authors intended to highlight that the dataset is not only extensive but includes state-of-the art data (like oxygen profiles from TV guided sampling stations, as described in the paragraph above), which were not available when the first diagenetic model was published in 2001 and thus justifies the presentation of this new modelling work. We will change to '[. . .] comprehensive set of data obtained with state-of-the-art methodology[. . .]'

However, I would suggest to also include porewater iron as this variable is discussed multiple times throughout the manuscript.

- Porewater iron concentrations are all near or below detection limit and this is described in the text (Line 226).

Line 61-63: how deep is the layer where there is organic matter degradation? Does this mean there is no iron of sulfate reduction?

- There is iron(III) reduction deeper in the sediment as witnessed by the tan-green color change in the sediment. The specialty in the Peru Basin is that the reduced Fe is not released into the porewater (thus dissolved Fe remains below or near the detection limit) but is incorporated into the clay mineral phase. $SO_4$ reduction is not reached however. See Haeckel et al., 2001.

Line 64: which depth is the bioturbation depth approximately?

- One highlight of the paper is that the bioturbation depth is derived from radiometric data, which is presented in Figure 7 and discussed in the section 'Diagenetic reference model'. The bioturbation depth reaches at least 10 cm, at which point the bioturbation intensity rapidly declines until around 15 cm below the seafloor.

Line 65: "Oxygen is depleted at a sediment depth of 5 to 15 cm, " Do you mean in general for deep see sediments? Then add a reference.

- No, this statement is located in the section 'Biogeochemistry of the Peru Basin' and is thus valid for the Peru Basin and taken from the references that are introduced three sentences before. As discussed for a comment further up, we will better clarify the sources of information for this section.

Line 65: "with the DEA region being on the deeper end". Add reference.

- Please see comment above

Line 66: "In the slightly shallower region of the Peru Basin seawater-derived nitrate and nitrate from oxic respiration is consumed within the bioturbated upper 20 cm of sediment". Add reference.

- Please see comment above

Line 70: It is a bit confusing to also refer to this layer as reaction layer as you also have a surface 'reaction layer'. I would use a different name iron-rich layer for example.

- We will remove 'reaction' for the description of the Fe(II)-rich layer to avoid confusion

with the surface 'reaction layer'.

Line 70: "formed during glacial times". Approximately how long ago?

- We will add: ',(corresponding to an age of at least 60 ka)'

Line 70: add reference to :" This reaction layer was formed during glacial times, when organic matter deposition was strongly increased and potential electron acceptors such as oxygen, nitrate and manganese oxide were exhausted closely below the sediment surface. "

- We will add the reference (König et al., 2001).

Line 79-85: Is this one of the main research questions? This is not described in the abstract.

- No. It is simply a justification of our presented modelling approach. As described above, we will move this paragraph to the description of the diagenetic model.

Line 89: "to artificially disturb the". This means without removing the reactive surface layer?

- Exactly, during the DISCOL experiment the sediment was ploughed but not removed. However, as part of this follow-up survey, a fresh disturbance was created with an epibenthic sledge, which removed the upper sediment layer and piled it up at the side of the track. Because the difference between 'ploughing' and 'removing' is indeed important, we highlighted the difference in the section 'Impact characterisation'.

Line 92: what does less disturbed mean. The sediment depth of disturbance?

- The effects of disturbance are manifold and include sediment mixing, displacement and blanketing with resettling sediment. 'Less disturbed' would for example be the description of an area that was affected only by a thin layer of resettling sediment. The gradient of disturbance plays an important role for this paper and is described in more detail in the impact characterization.

Line 97: I would give this information directly when you describe "artificially distur-bance".

- We will add to the description of the DISCOL experiment, where we first mention the artificial disturbance in the methods section: 'The DISCOL experiment was designed to artificially disturb the surface sediment layer by mixing and removing nodules from the surface with a specially designed device, the so-called plough-harrow (Thiel and Schriever, 90 1990).' This way it is clarified what kind of disturbance we mention here and can later be better contrasted to the disturbance created by the epibenthic sledge as described in Line 97.

Line 115: 20 mictoliter of HCl was added to how much sample? 1 mL?

- Yes, 1 mL. We will add this to the text: ' for 1 mL porewater'.

Line 116: "Additional sediment samples ". Do you mean a separate sediment core or the sediment leftovers after porewater extarction?

- No, these variables were measured from the same core and sediment intervals. For porosity a few ml of wet sediment were taken, POC was measured from this subsample as well, and part of the squeeze cake after porewater extraction was used for radio-metric analysis.

Line 119: you also show measured manganese data so you should add this to the list.

- We do not show solid manganese data and dissolved manganese data is mentioned as part of the sentence '[. . .] for analysis of metal cations [. . .]' in Line 116.

Line 119-124: Can you add the sample sizes to this paragraph or add this in Table 2.

- We will add the following sentence to Line 120: '1.3 mL of freshly extracted untreated pore water was diluted 3-fold before analysis.'

Line 154: Can you add equation numbers to all equations? That makes it easier to refer to them in the text.

- We will add equation number to all equations in the text.

Line 176: shortly describe how the gridsize is distributed.

- We will add to the existing information in brackets the following information: '(1D uneven grid with a sigmoidal distribution ranging from 0.1 cm at the surface and 1 cm at the lower boundary)

Line 178: describe that you use organic matter with different reactivities (Multi G approach) and give references for this approach (Jørgensen, 1978;Westrich and Berner, 1984; Middelburg, 1989).

- We will add the following sentence to Line 180: 'To account for different reactivities of the various organic matter phases, we apply a 3G model (Jørgensen, 1978;Westrich and Berner, 1984; Middelburg, 1989) allowing for a labile, moderately reactive and a refractory phase.

Line 195: I do not see a sharp drop in porosity in figure 5.

- Porosity decreases from 0.94 to 0.86 within the upper 10 cm and relative to the remaining core section, where porosity changes are only minimal, this decrease is indeed strong. We will replace 'sharp' with 'significant'.

Line 195: You should also give the reference to the figure.

- We will add a reference to the Figure.

Line 195: "and is rich in manganese". Can you add a concentration to this statement, or a range of concentrations.

- We will add '1- 2 wt%' to the reference.

Line 200: "Organic carbon content in the DEA region oscillates about a mean value of 0.5 to 0.75 wt% in the upper 50 cm,". refer to figure.

- We will add a reference to the figure

Line 202: "With depth the organic carbon content decreases steadily to about 1 wt% at 10 m.b". Add figure reference. Maybe I'm not looking at the right figure but the organic carbon content in fig 3 decreases to almost 0 and is never above 1 wt%.

- Yes, this is a typological mistake, it should be 0.1 wt%!

Line 203-207: This part seems more suitable to the discussion. I think it is better to first discuss the data you show in figures 3-6.

- We have divided the results section into (i) the description of the sediment and (ii) the description of the porewater. In the description of the sediment, sediment properties as well as composition of the sediment is included. Manganese nodules were found in the sediment core and their existence has to be mentioned here as well. We do not further discuss the implications of buried manganese nodules in this paper, because it is irrelevant for the research aims at hand. The authors would thus prefer to mention their existence only in the context of the core description.

Line 208-210: This is discussion, focus on presenting the data first.

- As described in a previous response, we believe that the paper benefits from including well established zones of organic matter degradation processes in the description of the results.

Line 217: "Dissolved manganese is strongly redox sensitive and is absent in the upper oxic zone." Add reference to such a statement.

- We will add a reference to Figure 4.

Line 226-228: Where can I see this Fe data? If you don't want to add this to the manuscript you can add it in an appendix.

- As mentioned in a response above. The visualization of data that is below the detection limit is fruitless and a statement in the text should suffice. We will add the detection limit of 1 $\mu$mol/L to the text to clarify the upper boundary of dissolved Fe.

Lines 231-235: I would add this part to the methods.

- The method section already includes all relevant information on the model. Keeping in mind that the first aim of this paper is an updated diagenetic model, the results section 'Diagenetic reference model' should include, in addition to the biogeochemical model results, a brief report on the success of this updated geochemical model. The introductory paragraph does exactly that; it states (i) that the 3G model allowed for a better model fit of the organic matter phase and (ii) that we were able to update information on bioturbation based on the new radiometric model presented in this paper. We would thus like to keep this paragraph in the results section.

Line 238: There should be a dot after inhomogeneities, Not a comma.

- This will be amended.

Line 241: Can you further explain why you choose these sites?

- We will add to Line 241: 'A satisfactory background model fit to undisturbed reference sites could be achieved [. . .]'

Line 248: "which suggests that secondary reactions, such as a semi-labile Fe(II)- oxygen reaction front, additionally shape the oxygen profile:" Why is this reaction not added in the model?

- As mentioned in the following sentence, running the background model as a transient state adds a complexity to the model that we are not able to parameterize. Long term changes in organic matter input likely create changes in the oxygen penetration depth, which in turn will be buffered (only at first!) by the Fe(II)-O2 reaction layer. The fact that the oxygen profile is less curved as we expect, can indeed be an indicator, that organic matter input is slowly waning at present, allowing the oxygen penetration depth to gradually migrate downwards, where it burns down the Fe(II)-O2 layer, creating this more linear profile. Trying to parameterize this transient state will not benefit the prognostic impact simulation and will not benefit the paper.

Line 260: "relevant information" is a bit vague.

- We will delete 'relevant'.

Line 264-266: Is this data compared to the reference stations?

- Yes, we will add 'compared to the reference sites'.

Line 268: "signs of bioturbation are lacking". how would you see these and are these sign visible at the reference site?

- The physical expression of bioturbation, or lack thereof, in the sediment core is described in detail in the referenced paper. We will add more information to the text: ' [. . .] physical signs of bioturbation (recent bioturbation channels connected to the surface) are lacking [. . .].'

Line 278: Can you give macrofaunal density (add a number).

- 1) The text says Megafauna – indeed as the reviewer suggests with his/her question, this should be macrofauna. 2) The observation in is based on 3D computed tomography scans – not on macrofauna densities. Specifically, the Vonnahme et al. analyzed whether the observed bioturbation channels extended all the way to the sediment surface (i.e., they are in use/recent) or not (i.e. they are likely abandoned/relict). Some likely recent bioturbation channels have been observed in the depressions created by the plough harrow (and to a lesser extent in the elevations/ridges). In the areas where deeper sediments were exposed at the surface, recent channels were missing. These observations have not been quantified by Vonnahme et al. (e.g., in terms of burrows / mˆ-2)

Line 292-295: These lines are repetition. This is already discussed in the introduction.

- We will delete the introductory sentences: 'Oxygen is essential to all multicellular life and represents the electron acceptor with the highest energy yield. It exerts an important control on the sediment redox zonation and associated reactions (most importantly the oxic respiration of organic matter), which in turn control the distribution of other nutrients, such as ammonium and nitrate.'

Lines 300-304: For me it is difficult to understand how these changes were modeled. Can this be explained in the method section in more detail?

- We will add to the methods section in Line 182 a paragraph on the transient model: 'Initial profiles of the transient simulations are based on the steady state background model which were adjusted for the modelled impact type: (i) in the case of 'sediment removal', the top 10 cm of the background profiles were cut off while maintaining bottom water concentrations and organic matter flux for the upper boundary conditions. For the 'sediment mixing' case, the oxygen concentrations of the upper 10 cm of the sediment were set to bottom water values, imitating an oxygen flooding event. Transient models were augmented by an additional Fe(II) profile - an oxygen reaction layer (Figure 2) - as justified by the surprisingly constant oxygen penetration depth at the reference and all disturbed sites. This Fe(II) reaction layer is crucial in the prognostic simulations which would otherwise predict that oxygen rapidly diffuses into deeper sediments after a disturbance event. It is assumed that bioturbation is inhibited immediately after the impact with a linear increase to undisturbed reference bioturbation intensity within 100 to 200 years. And we will delete the sentences of Lines 300-304.

Line 305: "It is assumed that bioturbation is inhibited immediately after the impact with a linear increase to undisturbed reference bioturbation intensity within 100 to 200 years". Where is this assumption based on, can you give a reference?

- We will add to the sentence '[...], which is in line with macrofaunal abundance data presented in Stratman et al., 2018.'

Line 311: "Upper centimetres". Give a number.

- We will add: 'restricted to the upper 10 centimetres'

Line 321: For me it is quite surprising that there is such a large difference in such a

short period.

- Yes indeed, the bioturbation has a surprisingly strong influence on the redox zonation of the shallow sediment. If labile organic matter is only degraded right at the surface, where oxygen is easily replenished, oxygen can migrate much deeper into the sediment. If however, through bioturbation labile organic matter is transported deeper into the sediment, oxic respiration also occurs at sediment depth where oxygen is only replenished by diffusion which is much less efficient and thus downward oxygen migration is much more limited. This is why we can notice a difference if bioturbation recovery takes 200 years instead of 100 years.

Line 346: "overall surface fluxes vary". Where can I see the all the fluxes besides oxygen?

- As mentioned in a previous response, we will add nitrate and ammonium fluxes to Figure 12.

Line 367 (Discussion): first paragraph does not give any new information or does discuss the results found in this study. The discussion reads as a summary/conclusion.

- The first paragraph of the discussion is the only place where we describe in brief the balanced interplay between the physical, chemical and biological factors that shape the benthic ecosystem. This information is vital when discussing the ecosystem recovery after an impact that affects all ecosystem compartments. While the information therein is not novel to this study, to highlight the essential feedbacks between microbial/macrofaunal abundances and geochemical fluxes is the substance of the paper and is reflected in the data presented as well as in the modelling.

Line 370: "The associated microbial growth". Add references

- The transfer of organic matter to microbial biomass has been demonstrated and quantified by pulse chase experiments with stable isotope-labeled organic matter in deep waters, including Pacific nodule provinces. Several experimental as well as modeling

studies also show that this organic matter is subsequently used by metazoans. We will add references to the text: '[...] The degradation of organic matter is microbially catalyzed and in turn provides nutrients to the microbial community (e.g., Witte et al. 2003a, Moodley et al., 2005, Stratmann et al., 2018, Sweetman et al., 2019). The associated microbial growth provides biomass that sustains meio and macrofauna and forms an important component of the benthic food web (Witte et al., 2003b, van Oevelen et al., 2006, 2011). [...]'

Moodley, L., Middelburg, J. J., Soetaert, K., Boschker, H. T. S., Herman, P. M. J., and Heip, C. H. R. (2005) Similar rapid response to phytodetritus deposition in shallow and deep-sea sediments, J. Mar. Res., 63: 457–469, https://doi.org/10.1357/0022240053693662 Stratmann, T., Mevenkamp, L., Sweetman, A. K., Vanreusel, A., van Oevelen, D. (2018) Has Phytodetritus Processing by an Abyssal Soft-Sediment Community Recovered 26 Years after an Experimental Disturbance? Front. Mar. Sci. 5:59, doi: 10.3389/fmars.2018.00059 Sweetman, A. K., Smith, C. R., Shulse, C. N., Maillot, B., Lindh, M., Church, M. J., ... Gooday, A. J. (2019). Key role of bacteria in the short-term cycling of carbon at the abyssal seafloor in a low particulate organic carbon flux region of the eastern Pacific Ocean. Limnology and Oceanography, 64(2), 694-713. https://doi.org/10.1002/lno.11069 Van Oevelen, D., Moodley, L., Soetaert, K., & Middelburg, J. J. (2006). The trophic significance of bacterial carbon in a marine intertidal sediment: Results of an in situ stable isotope labeling study. Limnology and Oceanography: 51: 2349-2359. van Oevelen, D., Bergmann, M., Soetaert, K., Bauerfeind, E., Hasemann, C., Klages, M., Schewe, I., Soltwedel, T., Budaeva, N. E. (2011) Carbon flows in the benthic food web at the deep-sea observatory HAUSGARTEN (Fram Strait), Deep-Sea Research I 58: 1069-1083 Witte, U., Aberle, N., Sand, M., and Wenzhoefer, F. (2003a) Rapid response of a deep-sea benthic community to POM enrichment: an in situ experimental study, Marine Ecology-Progress Series 251: 27-36 Witte, U., Wenzhoefer, F., Sommer, S., Boetius, A., Heinz, P., Aberle, N., Sand, M., Cremer, A., Abraham, W. R., Jorgensen, B. B., Pfannkuche, O. (2003b) In situ experimental evidence of the fate of a phytodetritus

pulse at the abyssal sea floor, Nature 424: 763-766

Line 379: "..infauna new material for additional degradation and microbial colonization, close the causal cycle." Add reference

- By means of stable isotope-labeled organic matter, several studies showed the downward mixing of organic matter and its use by microorganisms. We will add references to the text: '[. . .] Bioturbation plays thus a deciding factor in early-diagenetic, especially redox sensitive, processes. These are typically closely linked to the microbial functions and with the introduction of fresh and digested organic matter at depth by infauna new material for additional degradation and microbial colonization, close the causal cycle (Levin et al., 1997, Witte et al. 2003a, Middelburg, 2018) [. . .]'.

Levin, L., Blair, N., DeMaster, D., Plaia, G., Fornes, W., Martin, C., Thomas, C. (1997) Rapid subduction of organic matter by maldanid polychaetes on the North Carolina slope, J. Mar. Res., 55, 595-611 Middelburg, J. J.: Reviews and syntheses: to the bottom of carbon processing at the seafloor, Biogeosciences, 15, 413–427, https://doi.org/10.5194/bg-15-413-2018, 2018 Witte, U., Aberle, N., Sand, M., and Wenzhoefer, F. (2003a) Rapid response of a deep-sea benthic community to POM enrichment: an in situ experimental study, Mar. Ecol.-Prog. Ser., 251: 27-36

Lines 386-388:" through changing the physical characteristics of the sediment surface, i.e. by exposing more compacted sediments with increased shear strength, and removing manganese nodules as hard substrate habitats." Add reference.

- We will add the reference to this sentence: Grupe, B., H. J. Becker, and H. U. Oebius. 2001. Geotechnical and sedimentological investigations of deep-sea sediments from a manganese nodule field of the Peru Basin. Deep-Sea Res, Part II. 48: 3593–3608. doi:10.1016/S0967-0645(01)00058-3

Line 389: Where can I find the biological data?

- Throughout the text microbial abundances are quoted and set into the context of

[Figure]

biogeochemical processes. Also, Figure 12 presents referenced microbial and macro-faunal data.

Table 4: "Number of points in the numerical grid (uneven)". How are they distributed?

- As mentioned in a response to a previous comment, we will add a description of the uneven grid to the text: : '(1D uneven grid with a sigmoidal distribution ranging from 0.1 cm at the surface and 1 cm at the lower boundary)

"Rate constant for reaction of $Mn2+$ and $NO3$". This equation is not given in table 3

- This is an artefact from a previous manuscript version and should have been deleted. We found that this reaction does not play a significant role in our setting and set it to zero. This kinetic constant will be removed from the Table 3.

How do the rate constants (last column) compare with values used in the literature? Are they in the same range? I think it would be good to add a column which described the ranges of rate constants used in the literature (with references).

- Unfortunately, comparing kinetic constants is not very straight forward. We have tried to find suitable publications on comparable reaction rates for the organic matter degradation (e.g. Boudreau, 1996 in Computers and Geosciences, 22, p. 479 or Haeckel et al, 2001 in Deep-Sea Research II, 48, p. 3713 or Wang and Van Cappellen, 1996, in GCA, 60, p.2993), but different modelling approaches (e.g. 2G versus 3G) means that the kinetic constants cannot be compared. Also, secondary reactions strongly depend on the reaction network of the model and thus cannot be compared in a meaningful way.

There is no rate constant describing the third secondary reaction (oxidation of $Fe2+$)

- We will add the kinetic rate constant for the Fe(II)-O2 reaction to Table 4 (1e2 $mmol^{-1}$ $a^{-1}$)

"Monod constant for $SO42-$ reduction". This reaction is not in table 3. If you add this

reaction you should also add Fe reduction coupled to organic matter.

- SO4-reduction through organic matter degradation does not play a role at this site and we will thus remove it from Table 4.

Figure 8:

Total Corg. Not a very good fit. You could add more refractory OM deposition during the glacial periods.

- Porewater profiles in the upper half meter or so (Einstein-Smoluchowski relation) are defined by Holocene POC input and hence, it adds no further value to play around with past temporal changes of POC input during glacial times. Indeed, a better fit could be achieved if we would temporally correlate the refractory organic matter input with glacial/interglacial intervals. For the purpose of this impact orientated paper, the labile and semilabile organic matter fraction are by far more important and can be sufficiently well represented by our steady-state model. We will add in the description of the organic matter profile in Line 245: 'It should be noted that the geochemical state of the Peru basin is likely transient in nature arising from long-term (glacial/interglacial) as well as short-term (ENSO time scale) variations in the depositional flux of organic matter (König et al., 2001). This work focuses on modelling the biogeochemical processes in the upper half meter of the sediment, which are defined by the Holocene organic matter input, and are thus unaffected by long-term changes in the depositional flux.'

Also I would like to see how the three fractions are distributed with depth

- We will add another window to Figure 8 where the three different OM fractions will be plotted.

For me it is unclear why/how the selection of stations is made to fit the model. The stations are both inside and outside the DEA region.

- In addition to the three designated reference stations we have also included two other undisturbed sites that were sampled during the cruise to complete the picture of

the DEA area. This provides a better idea on the range of data scatter that can be expected from the DEA sites, especially with respect to the onset of dissolved manganese increase, for example. We will add this information to the Figure caption of Figure 8: 'Simulation of background biogeochemical processes and their effect on solute profiles in the DEA region. In addition to the three reference stations, two other undisturbed sites were included for comparison. See Table 4 for parameterization of the modelled profiles.'

Figure 10 FeII2: Why not show the steady state? I think it would be better to use the same type of line for impact in all the plots.

- We will add the Fe(II) steady state profile to Figure 10, which simply shows that the semi-labile Fe(II) reaction layer commences just below the O2 penetration depth, which is regulated by the present day organic matter input. Semi-labile Fe(II) that was present within the oxygenated zone has been 'burned down'.

---

## Author Comment (AC2) · 21 Jan 2020

Anonymous Referee #2 Haffert and co-authors present a comprehensive sedimentary geochemical dataset on an experimentally disturbed potential deep-sea mining area, called DISCOL. The paper reports an extensive set of downcore geochemical data (O2, organic C, nutrients) from both short (MUC, box corer) as well as long (GC) cores in the experimental site. Further, the study improves an existing diagenetic reaction-transport model with datadriven process optimisations. The authors then use these new data plus transient early diagenetic simulations to asses the short and long-term impact of sediment removal during (future) mining of polymetallic nodules. They find that the

removal of the surface labile organic carbon along with the nodules is the single most important driver of the establishment of a new geochemical regime in the disturbed areas. The primary strength of this work is a quite convincing set of predictions as to what will happen in the long run when these nodules will be extracted from the seabed. The transient simulations are very useful and well integrated to the data. Supported by an extensive and novel dataset and the improved modelling approach, the integrated methodology could be used in other seafloor resource extraction scenarios as well. There is no significant weakness in the manuscript. It is well laid out and well written. I only have a few suggestions for a minor revision of the existing manuscript:

Overall comment: How does the Fe(II)-rich clay layer can trap nitrate? The redox reaction between the two is not that well established, and wondering if a more complex cycle is present here, involving nitrogen intermediate species and a more complicated Fe(II)-Fe(III) cycle. I would propose that Figure 2 can be improved to clarify this, and both introduction (L84) and discussion parts can be expanded with a more detailed proposition of redox pathways.

- The Fe(II)-rich layer in the Peru basin was discussed in detail in previous publications (König et al., 1997; König et al., 1999; König et al., 2001). Intermediate complexes where not discussed in this context, instead changes in the deposition flux of organic matter has caused a 'redox pump mechanism'. Figure 2 summarizes the findings for the DEA region. It should be kept in mind that the NO3 burn down of the Fe(II)-rich layer, which occurs at present at much greater depth, does not play a role for the research question at hand. We would thus like to avoid a detailed discussion on the Fe(II)-Fe(III) cycle and focus on the shallow sediments directly affected by potential mining activities.

L27: JPI Oceans - with plural 'oceans', I think is the right acronym.

- Yes. We will correct this.

L54: the work 'untypically' can be removed without a significant change in the meaning

of the sentence.

- We will remove 'untypically' from the L54.

L58: 'availability positions' - perhaps could be re-phrased using 'order of the electron acceptors' or similar.

- The sentence is trying to say that the redox zones are controlled by the availability of the various electron acceptors. To clarify the sentence we will change it to: 'The reactions utilize different terminal electron acceptors in the order of decreasing free-energy production, namely oxygen, nitrate, manganese oxide, iron oxide and sulphate and their availability controls the position of the various redox zones in the sediment column.'

Section 1.1 - overall I find this section is more like a discussion, rather than introduction. As mentioned above, the introduction of a Fe(II)-NO3 redox pathway is a little bit out of place in this section. Besides, the readers might expect a following section of 1.2, since there is 1.1, bot there is no other subsection of the introduction. Please consider re-organizing the material in 1.1.

- In line with comments from Referee #1, we will restructure the information in the introduction. We will move the section, which intends to justify the introduction of a shallow Fe(II)-O2 reaction layer, to the description of the diagenetic model to Line 180:

L200-205 - interesting - did all cores include such buried nodule layers? I would strongly recommend to indicate the depth of these layers in the Figures 3-6 to be able to see directly in the figure if the nodule is impacting the geochemical profiles.

- About half of the gravity cores included buried nodules. We will add the depth of manganese nodules to the gravity core profiles in Figure 3.

---

## Author Response (AR2)

Dear Editorial Team,

thank you for inviting us to upload the files for the production process. We have amended the manuscript according the suggested technical corrections and a tracked version is appended to this message.

Best regards,

Laura Haffert and co-authors

**DISCOL experiment revisited: Assessing the temporal scale of deep-sea mining impacts on sediment biogeochemistry**

Laura Haffert1, Matthias Haeckel1, Henko de Stigter2 and Felix Janssen3

1 GEOMAR Helmholtz Centre for Ocean Research Kiel, Wischhofstrasse 1-3, 24148 Kiel, Germany

2 NIOZ - Royal Netherlands Institute for Sea Research and Utrecht University, P.O. Box 59, 1790 AB Den Burg
 - Texel, The Netherlands

3 HGF MPG Group for Deep Sea Ecology and Technology at the Max Planck Institute for marine Microbiology, Bremen and Alfred Wegener Institute Helmholtz Centre for Polar and Marine Research, Bremerhaven, Germany Max-Planck Institute for marine Microbiology, Bremen, Germany

**10**

5

Correspondence to: Laura Haffert (lhaffert@geomar.de)

Abstract. Deep-sea mining for polymetallic nodules is expected to have severe environmental impacts because in additionnot only to the nodules, but also benthic fauna as well as theand the upper reactive sediment layer is 
[revised manuscript text omitted]
                            | 7 04.578     | 88°27.781               | 4148             |
|          | 70 MOC 17      | DEA west resetted seatment plane                             | 7 04.400     | 88 21.118               | 4128             |
|          | 84 CC 3        | DEA black patch in sideseen soner                            | 7002 051     | 88°77 002'              | 4146             |
|          | 84 UC 5        | DEA block patch in sidescan sonar                            | 7 03.931     | 88 27.093               | 4140             |
|          | 74 MUC 20      | DEA black patch in sidescan sonar                            | 7 03.945     | 88 27.097               | 4150             |
|          | 100 CC 5       | DEA central/trougn                                           | 7004 2421    | 99907 440'              | 4151             |
|          | 100 GC 3       | DEA central                                                  | 7 04.342     | 88 27.442               | 4131             |
|          | 108 MUC 26     | DEA central                                                  | /*04.485     | 88-26.919               | 4169             |
| Microh   | abitats |                                                              |              |                         |                  |
|          |                | Reference                                                    |              |                         |                  |
|          | 34 MUC 6       | southern reference with nodules                              | 7°07.524'    | 88°27.031'              | 4162             |
|          | 80 MUC 22      | western reference with nodules                               | 7°04.542'    | 88°31.581'              | 4130             |
|          | 119 MUC 31     | eastern reference with(out) nodules
DEA outside tracks    | 7°06.033'    | 88°24.826'              | 4204             |
|          | 61 MUC 13      | DEA west outside plough track                                | 7°04.378'    | 88°27.781'              | 4148             |
|          | 146 ROV PC79   | DEA west 20 m off plough track                               | 7°04.4000'   | 88°27.8266'             | 4140             |
|          | 166 ROV PC70   | DEA east 20 m off plough track                               | 7°04,4585'   | 88°26.9240'             | 4143             |
|          | 229 MUC        | DEA south outside plough track                               | 7°04.6970'   | 88°27.3970'             | 4133             |
|          |                | DEA track furrow                                             | , 0.10770    | 20 21/29/0              |                  |
|          | 146 ROV PC77   | DEA west plough furrow                                       | 7°04.4110'   | 88°27.8363'             | 4139             |
|          | 166 ROV PC69   | DEA east plough furrow                                       | 7°04 4780'   | 88°26 9178'             | 4143             |
|          | 219 ROV PC75   | DEA south plough furrow                                      | 7°04 6930'   | 88°27 4540'             | 4155             |
|          | 217 KOV 1075   | DEA ridge                                                    | 1 07.0730    | 00 27.4340              | 7155             |
|          | 142 ROV PC33   | DEA west plough ridge                                        | 7°04 4094'   | 88°27 8330'             | 4139             |
|          | 162 DOV DC92   | DEA east plough ridge                                        | 7001 1026    | 88°76 0222'             | 4143             |

|         | 232 ROV PC64        | DEA south plough ridge        | 7°04.6890' | 88°27.4554' | 4156 |
|---------|---------------------|-------------------------------|------------|-------------|------|
|         |                     | DEA subsurface patch          |            |             |      |
|         | 142 ROV PC48        | DEA west subsurface patch     | 7°04.4113' | 88°27.8127' | 4140 |
|         | 169 ROV PC83        | DEA east subsurface patch     | 7°04.4808' | 88°26.9130' | 4144 |
|         | 219 ROV PC58        | DEA south subsurface patch    | 7°04.6930' | 88°27.4540' | 4155 |
|         |                     | EBS track                     |            |             |      |
|         | 202 ROV PC80        | DEA west inside EBS track     | 7°04.9533' | 88°28.1980' | 4150 |
|         | 202 ROV PC18        | DEA west side pile EBS track  | 7°04.9609' | 88°28.1907' | 4150 |
|         | 211 ROV PC52        | DEA west rim inside EBS track | 7°04.9581' | 88°28.1909' | 4150 |
|         | 211 ROV PC73        | DEA west outside of EBS track | 7°04.9669' | 88°28.1929' | 4150 |
| In situ | oxygen profiles     |                               |            |             |      |
|         |                     | Reference                     |            |             |      |
|         | 158 LANDER-#1       | Reference South               | 7°7.4590'  | 88°26.9740' | 4155 |
|         |                     | Undisturbed DEA               |            |             |      |
|         | 176 ROV profiler #1 | DEA E outside track           | 7°4.4677'  | 88°26.9187' | 4102 |
|         | 169 ROV profiler #1 | DEA E outside track           | 7° 4.4563' | 88°26.9176' | 4102 |
|         | 213 ROV profiler #2 | EBS outside track             | 7°5.0220'  | 88°28.1526' | 4189 |
|         | 146 ROV profiler #2 | DEA W outside track           | 7°4.4000'  | 88°27.8274' | 4140 |
|         |                     | DEA plough tracks             |            |             |      |
|         | 176 ROV profiler #2 | Subsurface                    | 7°4.4762'  | 88°26.9190' | 4102 |
|         | 154 ROV profiler #2 | Subsurface                    | 7°4.4118'  | 88°27.8172' | 4101 |
|         | 166 ROV profiler #2 | Furrow                        | 7°4.4898'  | 88°26.9286' | 4104 |
|         | 169 ROV profiler #2 | Furrow                        | 7°4.4787'  | 88°26.9205' | 4104 |
|         |                     | EBS track                     |            |             |      |
|         | 202 ROV profiler #1 | Inside track                  | 7°4.9787'  | 88°28.1730' | 4189 |

| Parameter                    | Method                                                         | Error (detection limit) a                    |
|------------------------------|----------------------------------------------------------------|---------------------------------------------------------|
| NO 3 - | Spectrophotometer (as sulphanile-a-naphthylamide) b | (1 µmol l -1 )                               |
| NO 2 - | Spectrophotometer (as sulphanile-a-naphthylamide) b | $(1 \ \mu mol \ \Gamma^1)$                              |
| $\mathrm{NH_4}^+$            | Spectrophotometer (as indophenol blue)b                        | $(1 \ \mu mol \ l^{-1})$                                |
| Mn 2+             | ICP-AES                                                        | 5-10% (1 μmol l -1 )                         |
| $SO_4^{2-}$                  | Ion chromatography                                             | $0.8-1.2 \text{ mmol } l^{-1} (5 \text{ mmol } l^{-1})$ |
| $C_{org}$                    | CHN-Analyser c                                      | 0.04 wt%                                                |
| Alkalinity                   | Titration d                                         | 0.05 meq l -1                                |
| Porosity                     | Weight difference before and after drying of the sediment      | 0.02                                                    |

Table 2. Summary of analysed properties, analytical methods, estimated analytical errors, and detection limits

a Note. For some properties no analytical error could be determined because of few data points or concentrations close to the detection limit.

b Grasshoff et al. (1999).

c Welicky et al. (1983).

d Breland and Byrne (1993).

**Table 3.** Reaction stoichiometry and rate expressions of the organic mater  $((CH_2O)_a(NH_3)_b(H_3PO_4)_c)$  degradation and secondary redox reactions. All solute species are in concentrations of  $\frac{mmol}{L_{pw}}$  and all solid species in  $\frac{mmol}{L_{ds}}$ . All

reaction rate expressions are stated in the units of  $\frac{mmol}{L_{pw}a}$  via a conversion with  $F_{pw} = \left(\frac{1-\phi}{\phi}\right)$ .

| Reaction | Rate expression (mmol/Lpw/a) |
|----------|------------------------------|
|          |                              |

**Organic matter degradation**

ID

 $\begin{array}{l} (CH_2O)_a(NH_3)_b(H_3PO_4)_c + (a+2b)O_2 \\ + (b+2c)HCO_3^- \to (a+b+2c)\ CO_2 \\ + (b)NO_3^- + (c)HPO_4^{2-} \\ + (a+2b+2c)H_2O \\ (CH_2O)_a(NH_3)_b(H_3PO_4)_c + \left(\frac{4}{5}a + \frac{3}{5}b\right)NO_3^- \\ & \quad \Rightarrow \frac{1}{2}\left(\frac{4}{5}a + \frac{3}{5}b + b\right)N_2 + cHPO_4^{2-} \\ + \left(\frac{3}{5}a + \frac{6}{5}b + 2c\right)H_2O \\ & \quad + \left(\frac{1}{5}a - \frac{3}{5}b + 2c\right)CO_2 \\ & \quad + \left(\frac{4}{5}a + \frac{3}{5}b - 2c\right)HCO_3^- \\ (CH_2O)_a(NH_3)_b(H_3PO_4)_c + 2a\ MnO_2 + (3a+b-2c)CO_2 \\ & \quad + (a+b-2c)H_2O \\ & \quad \Rightarrow (4a+b-2c)HCO_3^- + 2a\ Mn^{2+} \\ & \quad + bNH_4^+ + cHPO_4^{2-} \end{array}$

$$\sum_{i=1,2,3} k_i \frac{Corg_i}{a} R_{02} F_{pw}$$

$$\sum_{i=1,2,3} k_i \frac{Corg_i}{a} R_{NO3} F_{pw}$$

$$\sum_{i=1,2,3} k_i \, \frac{Corg_i}{a} \, R_{MnO2} \, F_{pw}$$

**Secondary redox reactions**

$$\begin{aligned} NH_4^+ + 2O_2 + 2HCO_3^- &\rightarrow NO_3^- + 2CO_2 + 3H_2O \\ 2Mn^{2+} + O_2 + 4HCO_3^- &\rightarrow 2MnO_2 + 4CO_2 + 2H_2O \\ 4Fe(II) + O_2 + 2H_2O + 4CO_2 &\rightarrow 4Fe(III) + 4HCO_3^- \end{aligned}$$

**Monod expressions**

[revised manuscript text omitted]